



**Formation and Evolution of Secondary Organic Aerosol Derived from Urban Lifestyle Sources:**
**Vehicle Exhaust and Cooking Emission**
Zirui Zhang[§,1], Wenfei Zhu[§,1], Min Hu[*,1,2,5], Kefan Liu[1], Hui Wang[1], Rongzhi Tang[1], Ruizhe Shen[1], Ying Yu[1], Rui Tan[1], Kai
Song[1], Yuanju Li[1], Wenbin Zhang[3], Zhou Zhang[3], Hongming Xu[3], Shijin Shuai[3], Shuangde Li[4], Yunfa Chen[4], Jiayun Li[6], Yuesi
Wang[6], Song Guo[1]
[1]State Key Joint Laboratory of Environmental Simulation and Pollution Control, International Joint Laboratory for Regional
Pollution Control, Ministry of Education (IJRC), College of Environmental Sciences and Engineering, Peking University,
Beijing 100871, China
[2]Collaborative Innovation Center of Atmospheric Environment and Equipment Technology, Nanjing University of
Information Science & Technology, Nanjing 210044, China P. R.
[3]State Key Laboratory of Automotive Safety and Energy, Tsinghua University, Beijing 100084, China
[4]State Key Laboratory of Multiphase Complex Systems, Institute of Process Engineering, Chinese Academy of Sciences,
Beijing 100190, China
[5]Beijing Innovation Center for Engineering Sciences and Advanced Technology, Peking University, Beijing 100871, China
[6]State Key Laboratory of Atmospheric Boundary Layer Physics and Atmospheric Chemistry (LAPC), Institute of Atmospheric
Physics, Chinese Academy of Sciences, Beijing 100029, China
[§]These authors contributed equally to this work.
*Correspondence to: Min Hu (minhu@pku.edu.cn).*
**ABSTRACT**

21         Both vehicle exhaust and cooking emission are closely related to the daily life of city dwellers, which are considered as

major sources of urban secondary organic aerosol (SOA). Here, we defined the SOA derived from vehicle exhaust and cooking
emission as "Urban Lifestyle SOA", and simulated their formation using a Gothenburg potential aerosol mass reactor (Go:
PAM). After samples had been aged under 0.3-5.5 days of equivalent photochemical age, these two urban lifestyle SOA
showed markedly distinct features in SOA mass growth potentials, oxidation pathways and mass spectra. The SOA/POA mass
ratios of vehicle groups (107) were 44 times larger than those of cooking groups (2.38) at about 2 days of equivalent
photochemical age. It reveals that organics from vehicle may undergo the alcohol/peroxide and carboxylic acid oxidation
pathway to produce abundant less/more oxidized oxygenated OA (LO-OOA and MO-OOA), and only a few primary
hydrocarbon-like organic aerosol (HOA) remains unaged. In contrast, organics from cooking may undergo the
alcohol/peroxide oxidation pathway to produce moderate LO-OOA, and comparable primary cooking organic aerosol (COA)
remains unaged. Our findings provide an insight into atmospheric contributions and chemical evolutions for urban lifestyle
SOA, which would greatly influence the air quality and health risk assessments in urban areas.





## 1. Introduction

Organic aerosol (OA) contributes 20-90% of submicron aerosols in mass (Jimenez et al., 2009;Zhang et al., 2011), and its fraction in urban areas is higher than that in suburban or background (Zhou et al., 2020). The OA could be divided into primary organic aerosol (POA) and secondary organic aerosol (SOA). POA is directly emitted into ambient air through coal combustion, biomass burning, vehicle exhaust, cooking procedure and so forth (Jimenez et al., 2009;Zhang et al., 2011;Zhou et al., 2020). SOA is formed via the oxidation of gas-phase organics and the distribution between gas and particle phase (Donahue et al., 2009). Significant SOA formation has been observed in several urban areas, but model failed to simulate this phenomenon accurately (Matsui et al., 2009;Kleinman et al., 2008;Volkamer et al., 2006;de Gouw et al., 2008). This discrepancy may attribute to the limited knowledge about the sources and characteristics of urban SOA.

Over the past decades, megacities have already been widespread in developed regions, and rapid urbanizations have been sweeping across the globe especially in developing areas (Zhang et al., 2015). An increasing number of people tend to live in the urban for their livelihood, where they suffer from serious air pollution simultaneously from urban lifestyle sources typically involving vehicle and cooking fumes (An et al., 2019;Zhang et al., 2015;Chan and Yao, 2008;Guo et al., 2014;Guo et al., 2020). For instance, polycyclic aromatic hydrocarbons (PAHs) are important carcinogens coming from vehicle and cooking, which can cause severe lung cancer (Seow et al., 2000;Kim et al., 2015;Zhong et al., 1999). After PAHs are emitted to ambient air, they would be oxidized, distributed into particle phase and finally become the part of POA or SOA, thus adding unknown deviations on health risk assessments (Masuda et al., 2020).

Vehicle and cooking emissions are important sources of OA in urban areas (Rogge et al., 1991;Rogge et al., 1993;Hu et al., 2015;Hallquist et al., 2016;Crippa et al., 2013;Mohr et al., 2012;Guo et al., 2013;Guo et al., 2012), take several megacities for example, in London and Manchester, these two lifestyle sources contributed 50% and 54% of OA in average (Allan et al., 2010). In addition, the vehicle itself could even contribute 62% of OA mass in rush hour of New York City (Sun et al., 2012). As for OA source appointments in Paris, vehicle and cooking contributed maximum 46-50% of OA (Crippa et al., 2013). According to seasonal observations in Beijing, there were at least 30% of OA coming from vehicle and cooking emissions (Hu et al., 2017). Briefly, these two urban lifestyle sources are closely related to the daily life of city residents and could account for 20-60% of ambient OA mass in urban areas when only considering their contributions to POA (Allan et al., 2010;Sun et al., 2011;Ge et al., 2012;Sun et al., 2012;Lee et al., 2015;Hu et al., 2017). Furthermore, the model speculated that vehicle and cooking emissions might even contribute over 90% of SOA in downtown Los Angeles by applying hypothetical parameters with a certain degree of uncertainty (Hayes et al., 2015). Therefore, vehicle and cooking are momentous sources of both POA and SOA in urban areas, and could be defined as "Urban Lifestyle Source of OA".

As is well-known, large amounts of volatile, semi-volatile and intermediate-volatility organic compounds (VOCs, SVOCs and IVOCs, respectively) are emitted from vehicle and cooking sources, leading to largely potential SOA productions (Klein et al., 2016;Katragadda et al., 2010;Liu et al., 2017c;Tang et al., 2019;Zhao et al., 2015;Esmaeilirad and Hosseini, 2018;Zhao et al., 2017;Yu et al., 2020). Lab studies have investigated the formation of vehicle or cooking SOA using a smog





chamber or an oxidation flow reactor (OFR). On the one hand, some lab experiments have investigated the vehicle SOA based
on variables such as fuel types, engine types, operating conditions and so on (Deng et al., 2020;Suarez-Bertoa et al., 2015;Zhao
et al., 2015;Du et al., 2018). Several smog chamber results found that the mass loading of SOA exceeded POA when the
equivalent photochemical age was more than one day (Gordon et al., 2013;Chirico et al., 2010;Nordin et al., 2013). Besides,
OFR could simulate a higher OH exposure, and the peak SOA production occurred after 2-3 days of equivalent atmospheric
oxidation (Tkacik et al., 2014;Zhao et al., 2018;Timonen et al., 2017;Watne et al., 2018;Alanen et al., 2017). The mass spectra
of vehicle SOA showed both semi-volatile and low-volatility oxygenated organic aerosol (SV-OOA and LV-OOA) features
along with the growth of oxidation degree (Tkacik et al., 2014). On the other hand, only a few lab experiments have
investigated the cooking SOA based on simplified ingredients or a single cooking method, involving heated cooking oils (Liu
et al., 2017a;Liu et al., 2018), stir-frying spices (Liu et al., 2017b) ,charbroiled meat (Kaltsonoudis et al., 2017) and Chinese
cuisines (Zhang et al., 2020b). These lab experiments indicated that the characteristics of SOA are influenced by multiple
factors, such as cooking methods, fuels, cookers or ingredients. The mass ratios of POA and SOA derived from cooking are
comparable, and the mass spectra of SOA showed much more similarities with the ambient semi-volatile oxygenated OA (SV-
OOA) factors (Liu et al., 2018). Although these lab studies have provided important insights into the secondary formation of
vehicle and cooking SOA, significant uncertainties still exist. Nobody has compared the different natures generated from
these two urban lifestyle sources in detail, let alone pointed out their potentially different roles in the real atmosphere.

In this work, we have designed our vehicle and cooking lab experiments according to daily basis situations in urban areas

of China. For vehicle exhaust simulation, China V gasoline and three common operation conditions were chosen. For cooking
emission simulation, four prevalent Chinese domestic cooking types were evaluated. A Gothenburg potential aerosol mass
reactor (Go: PAM) was used as the oxidation system. All the fresh or aged OA was characterized in terms of mass growth
potentials, elemental ratios, oxidation pathways and mass spectra. The aged OA could be divided into POA and SOA. The
latter was defined as "Urban Lifestyle SOA" whose mass spectra would be compared with those of ambient SOA, like less-
oxidized oxygenated OA (LO-OOA) and more-oxidized oxygenated OA (MO-OOA) measured in urban areas of China. These
findings are aim to support for the estimation of these two urban lifestyle SOA in ambient air, conducing to the policy
formulation of pollution source control and health risk assessment of exposure to vehicle and cooking fumes.
**2. Material and Method**
**2.1 Experimental Setup**

The vehicle experiment was conducted from July to October in 2019, at Department of Automotive Engineering,

Tsinghua University. The cooking experiment was conducted from November 2019 to January 2020, at Langfang Branch,
Institute of Process Engineering, Chinese Academy of Sciences. The field study was deployed at the Institute of Atmospheric
Physics (IAP), Chinese Academy of Sciences (39°58′N; 116°22′E) in autumn and winter (Autumn: Oct. 1st, 2018 – Nov. 15th,
2018; Winter: Jan. 5th, 2019 – Jan. 31st, 2019) (Li et al., 2020a). The sample site is located in the south of Beitucheng West
Road and west of Beijing Chengde expressway in Beijing, which is a typical urban site affected by local emissions (Li et al.,



2020b).

The lab simulations of two urban lifestyle SOA were conducted with the same oxidation and measurement system. Tables

1-2 contain information of vehicle and cooking experiment conditions. The vehicle exhaust was emitted from a gasoline direct
engine (GDI) with China V gasoline (similar to Euro V) under three speeds (20, 40, 60 km/h), which represented the urban
road condition in China (Zhang et al., 2020a). For all experiments, the gasoline direct injection (GDI) engine ran in a single
room, its exhaust was drawn into pipeline and then entered the Go: PAM at a 30 fold dilution where aerosols and gases reacted
at a stable temperature and relative humidity. On the other hand, four kinds of domestic cuisines were cooked with liquefied
petroleum gas (LPG) in an iron wok, including deep-frying chicken, shallow-frying tofu, stir-frying cabbage and Kung Pao
chicken composed of cucumbers, peanuts and chicken. The cooking time and oil temperature were different due to the inherent
features of ingredients. For all experiments, the closed kitchen was full of fumes where the vision was blurred and the air was
choky after a long time of cooking process. Subsequently, the cooking fumes were drawn into pipeline from kitchen to lab
and then entered the Go: PAM at an 8 fold dilution where aerosols and gases reacted at a stable temperature and relative
humidity. Both vehicle and cooking fumes were diluted at a constant ratio by a Dekati Dilutor (e-Diluter, Dekati Ltd.). The
Go: PAM was able to produce high OH exposures using an ultraviolet lamp ($\lambda$=254 nm) in the presence of ozone and water
vapor, in order to simulate the photochemical oxidation in the atmosphere (Li et al., 2019a;Watne et al., 2018). Blank
experiments were separately designed in the presence of boiling water or dilution air under the same condition. The OA
concentrations of blank groups were far below those of experimental groups, which indicated the background values were
minor (Table S1). More details about experimental design and instruments can be found in Section S1.
**2.2 Measurements of the Gas and Particle Phase.**

Figure 1 presents the design of this lab simulation. The gases and aerosols were emitted from GDI room or kitchen, then

reacted and sampled in a lab. A high-resolution time-of-flight aerosol mass spectrometer (HR-ToF-AMS, Aerodyne Research
Inc.) was used to identify the chemical compositions of OA (Nash et al., 2006). Its time resolution was 2 min (precisely, 1
min for a mass sensitive V-mode, and 1 min for a high mass resolution W-mode). Two sets of scanning mobility particle sizers
(SMPS-1, Differential Mobility Analyzer, Electrostatic Classifier model 3080; Condensation Particle Counter model 3778;
SMPS-2, Differential Mobility Analyzer, Electrostatic Classifier model 3082; Condensation Particle Counter model 3772;
TSI Inc.) scanned every 2 min before and after Go: PAM individually to identify the size distribution and number
concentration of particles. The SMPS-1 determined the mass concentration of POA, while the SMPS-2 determined the mass
concentration of aged OA, and their mass difference could be regarded as the SOA. A $SO_2$ analyzer (Model 43i, Thermo
Electron Corp.) was used to measure the decay of $SO_2$ in offline adjustment. A $CO_2$ analyzer (Model 410i, Thermo Electron
Corp.) was used to reduce the $CO_2$ interference to organic fragments in mass spectra of HR-ToF-AMS. The particle densities
were measured through the determination of DMA-CPMA-CPC system (DMA-Differential Mobility Analyzer, Electrostatic
Classifier model 3080, TSI Inc.; CPMA- Centrifugal Particle Mass Analyzer, version 1.53, Cambustion Ltd.; CPC-
Condensation Particle Counter, Condensation Particle Counter model 3778, TSI Inc.). The POA (precisely, primary





Hydrocarbon-like OA, HOA, usually comes from vehicle exhaust; primary Cooking OA, COA) was regarded as the OA
measured before Go: PAM, or the OA measured after Go: PAM when the OH exposure was zero. The aged vehicle OA and
aged cooking OA were measured after Go: PAM under certain OH exposure. In order to prevent freshly warm gas from
condensing on the pipe wall, sampling pipes were equipped with heat insulation cotton and a temperature controller. Silicon
tubes were used to dry the emissions before they entered measuring instruments. Prior to each experiment, all pipelines and
the Go: PAM chamber were continuously flushed with purified dry air until the concentrations of gases and particles were
minimal.
**2.3  Data Analysis.**
**2.3.1 HR-Tof-AMS Data**

The SQUIRREL 1.57 and PIKA 1.16 written in IGOR (Wavemetrics Incorporation, USA) were used to analyze the HR-

ToF-AMS data including mass concentrations, elemental ratios, ion fragments and mass spectra. The ionization efficiency
(IE), relative ionization efficiency (RIE) and collection efficiency (CE) were determined individually before data processing.
The 300 nm ammonium nitrate particles were applied for converting the instrument signals to actual mass concentrations
(Jayne et al., 2000;Drewnick et al., 2005). A default value (1.4) of relative ionization efficiency (RIE) of OA was adopted.
Another synchronous SMPS-2 was used to correct the collection efficiency (CE) of HR-ToF-AMS by comparing their mass
concentrations (Gordon et al., 2014a). In order to separate the POA and SOA from aged OA, the mass spectra were resolved
by positive matrix factorization (PMF) analysis (Ulbrich et al., 2009).
**2.3.2 Determination and Evaluation of Oxidation Conditions in Go: PAM**

The Go: PAM conditions for vehicle and cooking experiments can be seen in Tables 3-4. The OH exposures and

corresponding photochemical ages in Go: PAM were calculated through an offline adjustment based on the decay of SO₂ (Lambe
et al., 2011). As shown in equation (1), $K_{OH\text{-}SO_2}$ is the reaction rate constant of OH radical and SO₂ ($9.0\times10^{-13}$ molecule$^{-1}\cdot$cm$^3\cdot$s$^{-1}$).
The SO₂, f and SO₂, i are the SO₂ concentrations (ppb) under the conditions of UV lamp on or off respectively. The
photochemical age (days) can be calculated in equation (2) when assuming the OH concentration is $1.5\times10^6$ molecules·cm$^{-3}$
in the atmosphere (Mao et al., 2009).
$$OH\ exposure = \frac{-1}{K_{OH\text{-}SO_2}} \times ln(\frac{SO_{2,f}}{SO_{2,i}}) \qquad (1)$$
$$Photochemical\ age = \frac{OH\ exposure}{24\times3600\times1.5\times10^6} \qquad (2)$$

Except for the off-line calibration based on the decay of SO₂, a flow reactor exposure estimator was also used in this

study (Peng et al., 2016). The OH exposures calculated by both methods showed a good correlation (Figure S1&S2). This
estimator could also evaluate the potential non-OH reactions in flow reactor such as the photolysis of VOCs, the reactions
with O($^1$D), O($^3$P) and O₃. Our results showed that non-OH reactions were not significant except for the photolysis of
acetylacetone. But there is no acetylacetone from vehicle exhaust or cooking emission according to our measurements and
previous studies. The acetylacetone was usually considered as a kind of VOCs emitted from industrial production (Ji et al.,





2020). Therefore, its potential photolysis wouldn't take place during our cooking conditions, and OH reactions still played
the dominant role. Overall, our Go: PAM could reasonably simulate the oxidation process of cooking OA in ambient.

Furthermore, the external OH reactivity and OH exposure were both influenced by external OH reactants, such as NOx

and VOCs during experiments. The NOx concentration was measured by a NO-NO2-NOx Analyzer (Model 42i, Thermo
Electron Corporation, USA). As for VOCs, we have divided them into 5 types including alkane, alkene, aromatic, O-VOCs
(Oxidized VOCs, mainly included aldehyde and ketone) and X-VOCs (halogenated-VOCs) using the measurement of GC-
MS (Gas Chromatography-Mass Spectrometry, GC-7890, MS-5977, Agilent Technologies Inc). The compounds with
relatively high proportion were regarded as surrogate species for each type of VOCs. The total concentrations of VOCs were
determined by a portable TVOC Analyzer (PGM-7340, RAE SYSTEMS). The external OH reactivities for different vehicle
experiments ($10.4\sim20.2$ s$^{-1}$) were all comparable to that of off-line calibration result ($15.8$ s$^{-1}$), and the external OH reactivities
for different cooking experiments ($21.7\sim25.7$ s$^{-1}$) were also comparable to that of off-line calibration result ($24.0$ s$^{-1}$). Besides,
the ratio of OH exposure calculated by the estimator to that calculated by the decay of $SO_2$ ranged from 83% to 119% for
vehicle experiments and 97% to 111% for cooking experiments, which means that our off-line OH exposure could be a
representative value to all experiments. The mixing and wall loss conditions have already met our experiment needs. Detailed
tests about mixing condition and wall loss of the Go: PAM have been conducted in previous work according to Li et al.(Li et
al., 2019a) and Watne et al (Watne et al., 2018), which could be found in Figure S3(a). In this study, we still corrected the wall
loss of particle in each size bin measured by two synchronous SMPS (two SMPS run before and after Go: PAM respectively).
**3. Result and Discussion**
**3.1 Formation Potential of the Urban Lifestyle SOA.**

As Figure 2 shows, the mass growth potentials of two urban lifestyle SOA were quite different. Although their SOA/POA

mass ratios both increased gradually through functionalization reactions and finally reached the peak after 2-3 days of
equivalent photochemical age (Kroll et al., 2009), the overall SOA mass growth potentials of vehicle SOA were far larger
than those of cooking SOA. When the equivalent photochemical age was near 2 days (1.7 days), the mass growth potentials
of vehicle SOA ranged from 83 to 150. In contrast, the mass growth potentials of cooking SOA only ranged from 1.8 to 3.2
at about 2.1 days. Even if there was still a slight growth trend for cooking SOA at the highest OH exposure, it surely exhibited
a much weaker mass growth potential on the whole compared with that of vehicle SOA. This significant distinction indicated
that the vehicle exhaust may contribute abundant SOA and relatively fewer POA, while cooking emission may produce
moderate POA and SOA in the atmosphere, which could attribute to their different types of gaseous precursors. For instance,
vehicle tended to generate large amounts of aromatics and cycloalkanes, which showed high rate constants of reaction with
OH and would lead to large SOA yields (Zhang et al., 2020a;Atkinson and Arey, 2003;Peng et al., 2017). By contrast, cooking
tended to emit much more unsaturated fatty acids that were tough to be oxidized even under high OH exposures (Zeng et al.,
2020;Nah et al., 2013). Interestingly, a similar phenomenon had been observed from an OFR simulation in the urban roadside
of Hongkong where potential SOA from motor vehicle exhaust was much larger than primary HOA, while potential SOA



from cooking emission was comparable to primary COA (Liu et al., 2019).
**3.2 Formation Pathway of the Urban Lifestyle SOA.**
As Figure 3 shows, the O:C molar ratios (O/C) of two urban lifestyle SOA were quite different. Although their oxidation
degrees both increased gradually and finally reached the peak after 2-3 days of equivalent photochemical age, the O/C values
of vehicle SOA were far larger than those of cooking SOA. When the equivalent photochemical age was 0.6 day, the O/C of
vehicle SOA was 0.4-0.5, resembling a kind of LO-OOA in ambient air. When the equivalent photochemical age was near 2
days (1.7 days), the O/C of vehicle SOA could reach 0.6, which was almost like a type of MO-OOA in the atmosphere. In
contrast, the O/C of cooking SOA only rose to 0.4 at 2.1 days, similar to a kind of LO-OOA. These distinct features of O/C
suggested that vehicle SOA was divided into LO-OOA and MO-OOA under different oxidation conditions, while the cooking
SOA was only composed of LO-OOA. This difference was probably related to their precursors. For example, vehicle emitted
large amounts of aromatics such as toluene, producing abundant SOA with a higher state of oxidation (Zhang et al.,
2020a;Suarez-Bertoa et al., 2015;Nordin et al., 2013;Liu et al., 2015;Deng et al., 2017). On the contrary, cooking generated
many unsaturated fatty acids such as oleic acid, which would remain unreacted under high OH exposures and thus retained
some features of fresh POA (Nah et al., 2013;Klein et al., 2016).
Figure 4 illustrates diverse oxidation pathways of various sources of OA in a Van Krevelen diagram (Heald et al.,
2010;Ng et al., 2011;Presto et al., 2014). The cooking groups fell along a line with a slope of -0.10 implying an
alcohol/peroxide pathway in forming SOA, while the vehicle groups fell along a line with a slope of -0.55 implying an
oxidation pathway between alcohol/peroxide and carboxylic acid reaction. Additionally, these two secondary evolution
properties are both different from those of biomass burning OA (slope≈-0.6) (Lim et al., 2019) and ambient OA (slope≈-1 to
-0.5) (Heald et al., 2010;Hu et al., 2017;Ng et al., 2011), indicating that these two urban lifestyles SOA may undergo distinct
oxidation pathways.
**3.3 Characteristics in Mass Spectra of the Urban Lifestyle SOA.**
As shown in Figure 5, m/z 43 ($f_{43}$) vs. m/z 44 ($f_{44}$) plot has been widely adopted to represent the oxidation process of
OA (Ng et al., 2010;Hennigan et al., 2011). Generally, $f_{43}$ and $f_{44}$ derive from oxygen-containing fragments, the former comes
from less oxidized components while the latter comes from more oxidized ones. The datasets of vehicle and cooking groups
apparently fell along in different regions and showed different variations in the plot. Almost all cooking OA displayed
relatively lower $f_{44}$ and higher $f_{43}$, and its $f_{43}$ and $f_{44}$ both increased slightly with the growing OH exposure, eventually
distributing in the LO-OOA region. In contrast, all vehicle OA displayed moderate $f_{43}$ and abundant $f_{44}$, and only its $f_{44}$
showed an obvious souring with the growing OH exposure, initially distributing in the LO-OOA region but finally spreading
near the MO-OOA region. These distinct evolutions of oxygen-containing fragments for two urban lifestyle SOA inferred
their intrinsic oxidation pathways and precursors. Vehicle might emit more easily oxidized aromatics, e.g. toluene and xylene,
while cooking might produce more hardly oxidized fatty acids such as palmitic acid and octadecanoic acid (Zhao et al.,
2007;Reyes-Villegas et al., 2018;Schauer et al., 2002;Zeng et al., 2020;Deng et al., 2017;Gordon et al., 2014b;Nah et al.,





2013;Zhang et al., 2020a).

Figure 6 and Table 5 depict mass spectra and prominent peaks of aged OA from two urban lifestyle sources which could

be used to deduce their inherent properties (Zhang et al., 2005;Kaltsonoudis et al., 2017;Liu et al., 2018;Chirico et al.,
2010;Nordin et al., 2013;Zhang et al., 2020b). The maximum SOA mass growth potentials of aged cooking SOA only ranged
from 1.9-3.2 implying a mixture of POA and SOA, so its mass spectra needed to be deeply resolved by PMF in order to
separate the POA and SOA (precisely, a kind of LO-OOA). However, those mass growth potentials of aged vehicle OA were
extremely high, suggesting that it was fully oxidized and almost composed of SOA. According to the O/C ratios, the vehicle
SOA under 0.6 day of photochemical age was defined as vehicle LO-OOA, while that under 2.9 days was regarded as vehicle
MO-OOA.

For average vehicle LO-OOA mass spectra, the prominent peaks were m/z 43 ($f_{43}$=0.133), 44 ($f_{44}$=0.077), 29 ($f_{29}$=0.076),

28 ($f_{28}$=0.066), 41 ($f_{41}$=0.051), 55 ($f_{55}$=0.043) dominated by $C_2H_3O^+$, $C_3H_7^+$, $CO_2^+$, $CHO^+$, $C_2H_5^+$, $CO^+$, $C_3H_5^+$, $C_3H_3O^+$ and
$C_4H_7^+$ respectively, while the prominent peaks of average vehicle MO-OOA were m/z 44 ($f_{43}$=0.146), 28 ($f_{44}$=0.134), 43
($f_{43}$=0.117), 29 ($f_{29}$=0.071), 45 ($f_{45}$=0.032), 27 ($f_{27}$=0.031) dominated by $CO_2^+$, $CO^+$, $C_2H_3O^+$, $CHO^+$, $C_2H_5^+$, $CHO_2^+$, $C_2H_5O^+$
and $C_2H_3^+$ respectively. Compared with vehicle SOA mass spectra from other studies (Table 5), our average GDI SOA (LO-
OOA and MO-OOA) illustrated more abundances of oxygen-containing ions than those of Gasoline SOA and Diesel SOA
simulated by a smog chamber with lower OH exposures (Chirico et al., 2010;Nordin et al., 2013).

For average cooking LO-OOA, it was less oxidized than those from vehicle groups, whose prominent peaks were m/z

43 ($f_{43}$=0.097), 44 ($f_{44}$=0.065), 29 ($f_{29}$=0.065), 41 ($f_{41}$=0.058), 55 ($f_{55}$=0.056), 28 ($f_{28}$=0.053) dominated by $C_2H_3O^+$, $C_3H_7^+$,
$CO_2^+$, $CHO^+$, $C_2H_5^+$, $C_3H_5^+$, $C_3H_3O^+$, $C_4H_7^+$ and $CO^+$ respectively. Compared with other cooking SOA mass spectra (Table
5), our average cooking LO-OOA had similar peaks with heated oil SOA, but was different from that meat charbroiling SOA
which displayed much more hydrocarbon-like features (Liu et al., 2018;Kaltsonoudis et al., 2017).
**3.4 Potential Chemical Evolution of Urban Lifestyle SOA in the Atmosphere.**

The AMS mass spectra indicated that the chemical evolution of urban lifestyle SOA in the Go: PAM might provide new

insights and references on those of ambient SOA observed in the atmosphere. Figure 7 plots the correlation coefficients
between the lab aged OA and ambient PMF-OA factors with growing photochemical ages (Li et al., 2020a). Table 6 exhibits
correlations of mass spectra between lab results and ambient PMF factors, where the aged lab cooking OA was divided into
POA and LO-OOA while the lab vehicle OA was divided into LO-OOA and MO-OOA.

For aged GDI OA in Figure 7, its average mass spectra still remained some ambient HOA features (pearson r=0.80)

under low photochemical age of 0.6 day with moderate hydrocarbon-like ions such as m/z 41 and 55, but it had already
reached the same oxidation degree of ambient LO-OOA (pearson r=0.81) with high O/C (0.46) and $f_{43}$ (0.133). After aging
in the Go: PAM, aged OA might finally become a kind of ambient MO-OOA (pearson r=0.97) at 5.1 days of photochemical
age. This evolution of GDI OA (from HOA to LO-OOA to MO-OOA) was similar to the result of a previous vehicle OA
simulation (from HOA to SV-OOA to LV-OOA) (Tkacik et al., 2014).





For aged cooking OA in Figure 7, although its correlations with ambient LO-OOA increased gradually from 0.56 to 0.73
along with the growing photochemical ages, its correlations with ambient COA kept a high level all the time (pearson r>0.81)
implying a mixture of POA and SOA due to some hardly oxidized compounds emitted from the cooking process. Therefore,
it is necessary to resolve aged cooking OA mass spectra deeply by PMF (Figures S4-S11) and then compared its lab PMF
results with ambient PMF factors. As Table 6 shows, the lab cooking POA was similar to ambient COA (pearson r=0.86) but
less likely to LO-OOA (pearson r= 0.46) or MO-OOA (pearson r=0.39). By contrast, the lab cooking LO-OOA displayed
many more ambient LO-OOA features (pearson r=0.76) and relatively fewer ambient COA characteristics than lab cooking
POA did. In short, these comparisons between lab and ambient results revealed that organics from these two urban lifestyle
sources might eventually form different SOA types in the real atmosphere.
**4. Conclusion**
In the present work, we define two urban lifestyle SOA in details and investigate their mass growth potentials, formation
pathways, mass spectra, and chemical evolutions comprehensively. At about 2 days of equivalent photochemical age, the
SOA/POA mass ratios of vehicle groups (107) were 44 times larger than those of cooking groups (2.38), and the O: C molar
ratios of vehicle groups (0.66) was about 2 times large as those of cooking groups (0.34). Besides, both vehicle and cooking
groups underwent alcohol/peroxide pathway to form LO-OOA, and the vehicle groups extra underwent carboxylic acid
pathway to form part of MO-OOA. Furthermore, the characteristic mass spectra of these two urban lifestyle SOA could
provide necessary references to estimate their mass fractions in ambient air, through a multilinear engine model (ME-2)
(Canonaco et al., 2013;Qin et al., 2017). This application would reduce the large gaps of total atmospheric contributions and
relevant environment effects for urban SOA, although remaining several uncertainties on SOA mass spectra due to missing
complex mixture conditions in the Go: PAM.
Although strict policies have been implemented to reduce primary particulate matter (PM) in urban areas. However,
secondary PM especially for the abundant and complicated SOA, is difficult to be restricted (Wu et al., 2017;Li et al., 2018).
According to our results, on the one hand, vehicle SOA might be a mixture of both LO-OOA and MO-OOA with high
secondary formation potential, so it would be better not only filter out the exhaust PM with Gasoline Particulate Filter (GPF),
but also reduce the gaseous precursors in order to restrict the secondary formation. On the other hand, cooking SOA might be
a kind of LO-OOA with relatively low secondary formation potential, so it could be enough to remove the gas and particle
emissions at the same level. In the future, these two urban lifestyle SOA may present increasing contributions in urban areas
especially in megacities with growing atmospheric oxidants (Li et al., 2019b;Wang et al., 2017;Li et al., 2020a;Li et al., 2020b),
but their investigations and further managements are far from sufficient, making it possible to become a greatly meaningful
research focus.







*Data availability.* The data provided in this paper can be obtained from the author upon request (minhu@pku.edu.cn).

*Supplement.* An independent supplement document is available.

*Authorship contributions.* Zirui Zhang: Investigation, Data curation, Methodology, Formal analysis, Writing - original draft, Writing - review & editing. Wenfei Zhu: Investigation, Data curation, Methodology, Formal analysis, Writing - review & editing. Min Hu: Project administration, Supervision, Funding acquisition, Writing - review & editing. Kefan Liu: Investigation, Data curation, Formal analysis. Hui Wang: Investigation, Data curation. Rongzhi Tang Investigation, Data curation. Ruizhe Shen: Investigation, Data curation. Ying Yu: Investigation, Data curation. Rui Tan: Investigation, Data curation. Kai Song: Investigation, Data curation. Yuanju Li: Investigation, Data curation. Wenbin Zhang: Investigation, Data curation. Zhou Zhang: Investigation, Data curation. Hongming Xu: Data curation. Shijin Shuai: Data curation. Shuangde Li: Data curation. Yunfa Chen: Data curation. Jiayun Li: Data curation. Yuesi Wang: Data curation. Song Guo: Project administration, Funding acquisition, Writing - review & editing.

Note: Zirui Zhang and Wenfei Zhu contributed equally to this work.

*Competing interests.* The authors declare that they have no known competing financial interests or personal relationships that could have appeared to influence the work reported in this paper.

*Acknowledgements.* Thanks to all authors from PKU who had directly participate in the main lab simulation. Thanks to all authors from THU and CAS who had provide necessary experiment sites, instruments and data supports.

*Financial support.* The research has been supported by the National Key R&D Program of China (2016YFC0202000), the National Natural Science Foundation of China (51636003, 91844301, 41977179, and 21677002), Beijing Municipal Science and Technology Commission (Z201100008220011), Open Research Fund of State Key Laboratory of Multiphase Complex Systems (MPCS-2019-D-09), and China Postdoctoral Science Foundation (2020M680242).



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





**Table 1.** Descriptions of vehicle exhaust and sampling procedures.

| Experiment | Revolving Speed | Torque | Sampling Time | Parallels | Particle Density | Fuel | Sampling Line Temerature |
|---|---|---|---|---|---|---|---|
| GDI 20 km/h | 1500 Hz | 16 N·m | 60 min | 3~5 | | | |
| GDI 40 km/h | 2000 Hz | 16 N·m | 70 min | 3~6 | 1.1~1.2 g/cm³ | Gasoline (China V, similar to Euro V) | 20~25°C |
| GDI 60 km/h | 1750 Hz | 32 N·m | 60 min | 3~5 | | | |


699





**Table 2.** Descriptions of cooking emission and sampling procedures.

| Experiment | Cooking Material | Oil Temperature | Total Cooking Time | Number of Dishes | Sampling Time | Parallels | Particle Density | Fuel & Cooware | Kitchen Volume | Sampling Line Temerature |
|---|---|---|---|---|---|---|---|---|---|---|
| Deep-fried Meat | 170 g chicken, 500 ml corn oil and a few condiments | 145~155°C | 66 min | 5 | 90 min | 3~8 | 1.11±0.02 g/cm³ | | | |
| Shallow-fried Tofu | 500 g tofu, 200 ml corn oil and a few condiments | 100~110°C | 64 min | 5 | 60 min | 3~5 | 1.04±0.03 g/cm³ | Liquefied petroleum gas (LPG) & iron wok | 78 m³ (5.6 m × 4 m × 3.5 m) | 20~25°C |
| Stir-fried Cabbage | 300 g cabbage, 40 ml corn oil and a few condiments | 95~105°C | 47 min | 5 | 58 min | 3~5 | 1.16±0.03 g/cm³ | | | |
| Kung Pao Chicken | 150 g chicken, 50 g ceanut, 50 g cucumber, 40 ml corn oil and a few condiments | Unmeasured[a] | 40 min | 5 | 60 min | 3~5 | 1.07±0.02 g/cm³ | | | |

[a]It is need to stir constantly, so the oil temperature was unstable.

089




**Table 3.** The Go: PAM condition for vehicle experiment.

| Experiment | $O_3$ concentration (ppbV) | OH Exposure[a] ($\times 10^{10}$ molecules·cm⁻³·s) | Photochemical Age (days, $[OH]=1.5\times10^6$ molecules·cm⁻³) | External OH reactivity of $SO_2$ during offline calibration ($S^{-1}$) | External OH reactivity of VOCs during experiment ($S^{-1}$) | Ratio of OH Exposure calculated[b] by an estimator to that calculated by the decay of $SO_2$[a] | Temperature & RH in Go :PAM | Basic Description of Go: PAM | Wall Loss |
|---|---|---|---|---|---|---|---|---|---|
| GDI 20 km/h | 624 | 7.79 | 0.6 | | | | | | |
| | 2367 | 21.4 | 1.7 | | | | | | |
| | 4433 | 37.4 | 2.9 | | 10.4 | 119% | | Voluem: 7.9 L. Flow rate: 4 L/min for sample air and 1 L/min for sheath gas. Residence time: 110 s. | The wall loss of particle have been adjusted in each size bin measured by two synchronous SMPS (two SMPS runned before and after Go: PAM respectively).The wall loss of gas phase is minor according to previous reseach. |
| | 6533 | 53.8 | 4.2 | | | | Temp: 19~22°C RH: 44-49% | | |
| | 8050 | 65.6 | 5.1 | 15.8 | | | | | |
| | 8701 | 70.6 | 5.5 | | | | | | |
| GDI 40 km/h | The same as 20 km/h experiments | | | | 20.2 | 83% | | | |
| GDI 60 km/h | The same as 20 km/h experiments | | | | 16.7 | 94% | | | |

[a] OH exposure was calculated based on the decay of $SO_2$.
[b] OH exposure for each ingredient was calculated based on the OFR estimator.



**Table 4.** The Go: PAM condition for cooking experiment.

| Experiment | $O_3$ concentration (ppbV) | OH Exposure[a] ($\times 10^{10}$ molecules·cm$^{-3}$·s) | Photochemical Age (days, [OH]=$1.5\times10^6$ molecules·cm$^{-3}$) | External OH reactivity of $SO_2$ during offline calibration (S$^{-1}$) | External OH reactivity of VOCs during experiment (S$^{-1}$) | Ratio of OH Exposure calculated by an estimator[b] to that calculated by the decay of $SO_2$[a] | Temperature & RH in Go: PAM | Basic Description of Go: PAM | Wall Loss |
|---|---|---|---|---|---|---|---|---|---|
| | - | 0 | 0.0 | | | | | | The wall loss of particle have been adjusted in each size bin measured by two synchronous SMPS (two SMPS runned before and after Go: PAM respectively).The wall loss of gas phase is minor according to previous reseach. |
| Deep-fried Chicken | 310 | 4.3 | 0.3 | | | | | | |
| | 1183 | 9.6 | 0.7 | | 25.7 | 97% | | | |
| | 2217 | 14.4 | 1.1 | | | | | Voluem: 7.9 L. Flow rate: 7 L/min for sample air and 3 L/min for sheath gas. Residence time: 55 s. | |
| | 3267 | 21.4 | 1.7 | | | | | | |
| | 4025 | 27.1 | 2.1 | | | | | | |
| Shallow-fried Tofu | The same as Meat experiments | | | 24.0 | 21.7 | 111% | Temp: 16~19°C RH: 18~23% | | |
| Stir-fried Cabbage | The same as Meat experiments | | | | 23.3 | 104% | | | |
| Kung Pao Chicken | The same as Meat experiments | | | | 23.6 | 103% | | | |

[a] OH exposure was calculated based on the decay of $SO_2$.
[b] OH exposure for each ingredient was calculated based on the OFR estimator.





**Table 5.** A summary of elemental ratios and dominant peaks among various SOA.

| Type | O/C | H/C | $f_{28}$ | $f_{29}$ | $f_{41}$ | $f_{43}$ | $f_{44}$ | $f_{55}$ | $f_{57}$ | Dominent Peaks (In deceding order) |
|---|---|---|---|---|---|---|---|---|---|---|
| GDI LO-OOA | 0.46 | 1.80 | 0.066 | 0.076 | 0.051 | 0.133 | 0.077 | 0.043 | 0.029 | m/z 43, 44, 29, 28, 41, 55 |
| GDI MO-OOA | 0.91 | 1.57 | 0.134 | 0.071 | 0.026 | 0.117 | 0.146 | 0.024 | 0.013 | m/z 44, 28, 43, 29, 45, 27 |
| Cooking LO-OOA | 0.36 | 1.92 | 0.053 | 0.065 | 0.058 | 0.097 | 0.065 | 0.056 | 0.046 | m/z 43, 44, 29, 41, 55, 28 |
| Heated oil SOA (Liu, 2018) | 0.38 | 1.53 | 0.070 | 0.087 | 0.067 | 0.078 | 0.067 | 0.053 | 0.023 | m/z 29, 43, 28, 44, 41, 55 |
| Meat charbroiling SOA (Kaltsonoudis, 2017) | 0.24 | 1.83 | 0.039 | 0.061 | 0.077 | 0.075 | 0.052 | 0.074 | 0.035 | m/z 41, 43, 55, 29, 27, 44 |
| Gasoline SOA (Nordin, 2013) | 0.40 | 1.38 | 0.122 | 0.032 | 0.031 | 0.094 | 0.129 | 0.019 | 0.008 | m/z 44, 28, 39, 27, 29, 41 |
| Disel SOA (Chirico, 2010) | 0.37 | 1.57 | 0.069 | 0.092 | 0.062 | 0.112 | 0.073 | 0.045 | 0.022 | m/z 43, 29, 44, 28, 41, 27 |

**Table 6.** Pearson correlations between lab OA and ambient OA mass spectra.

| Pearson Correlation ($\alpha=0.05$) | Ambient HOA | Ambient COA | Ambient LO-OOA | Ambient MO-OOA |
|---|---|---|---|---|
| Lab Cooking POA | 0.95 | **0.86** | 0.46 | 0.39 |
| Lab Cooking LO-OOA | 0.90 | **0.81** | **0.76** | **0.68** |
| Lab Vehicle LO-OOA | **0.80** | 0.71 | **0.81** | **0.73** |
| Lab Vehicle MO-OOA | 0.54 | 0.44 | **0.98** | **0.94** |





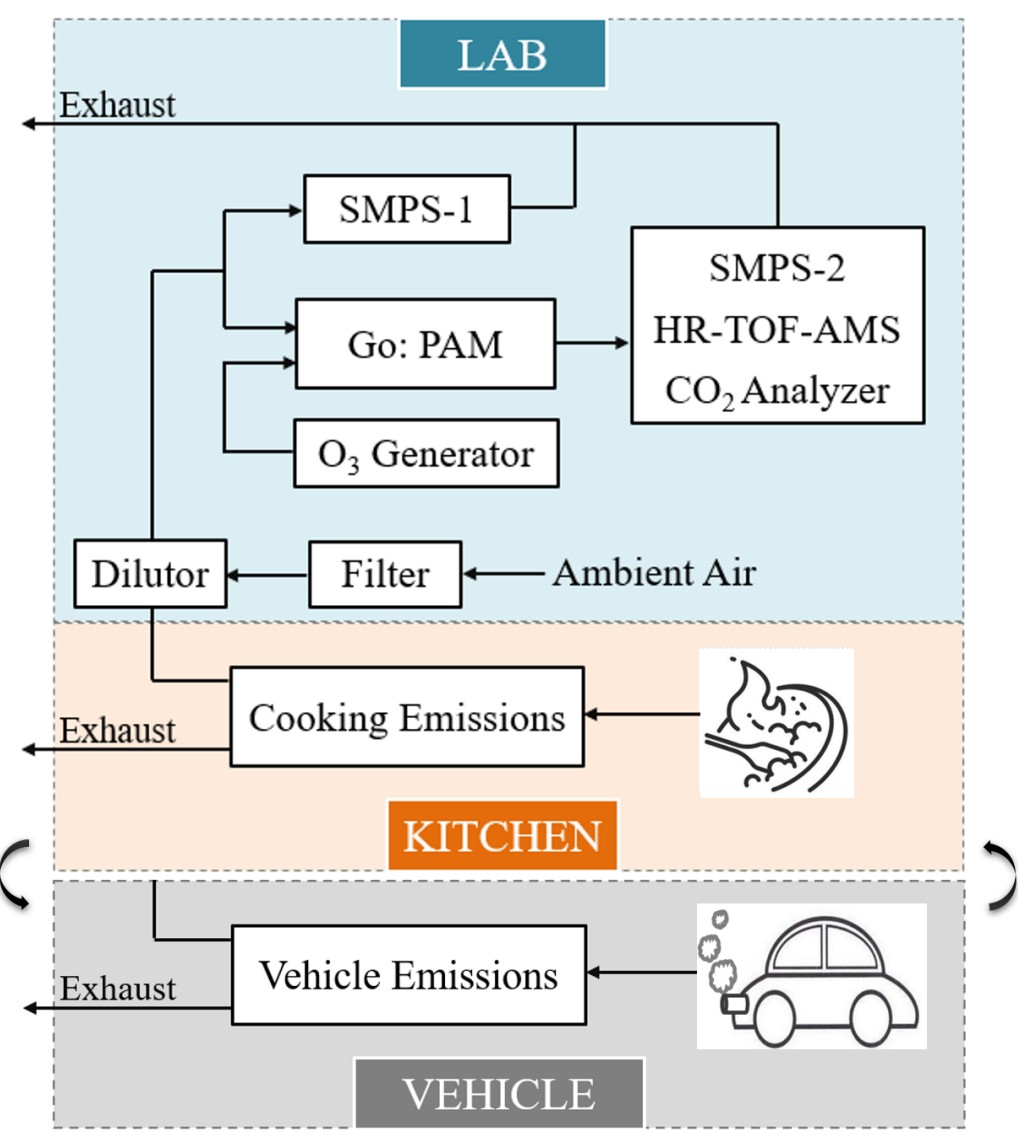


**Figure 1.** Schematic of experiment system.



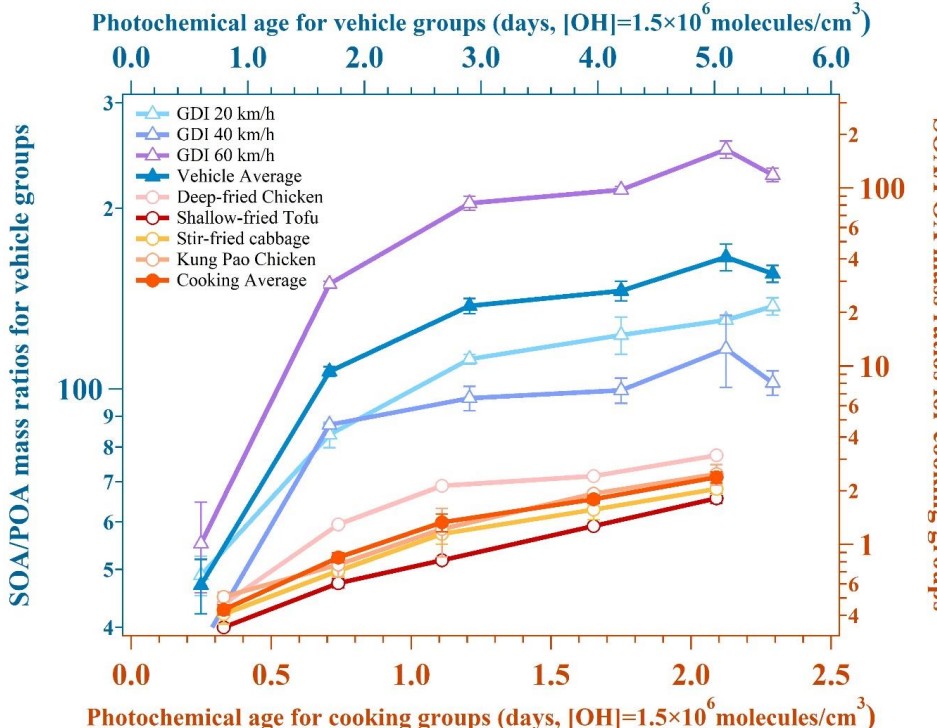


**Figure 2.** Mass growth potentials for two urban lifestyle SOA.

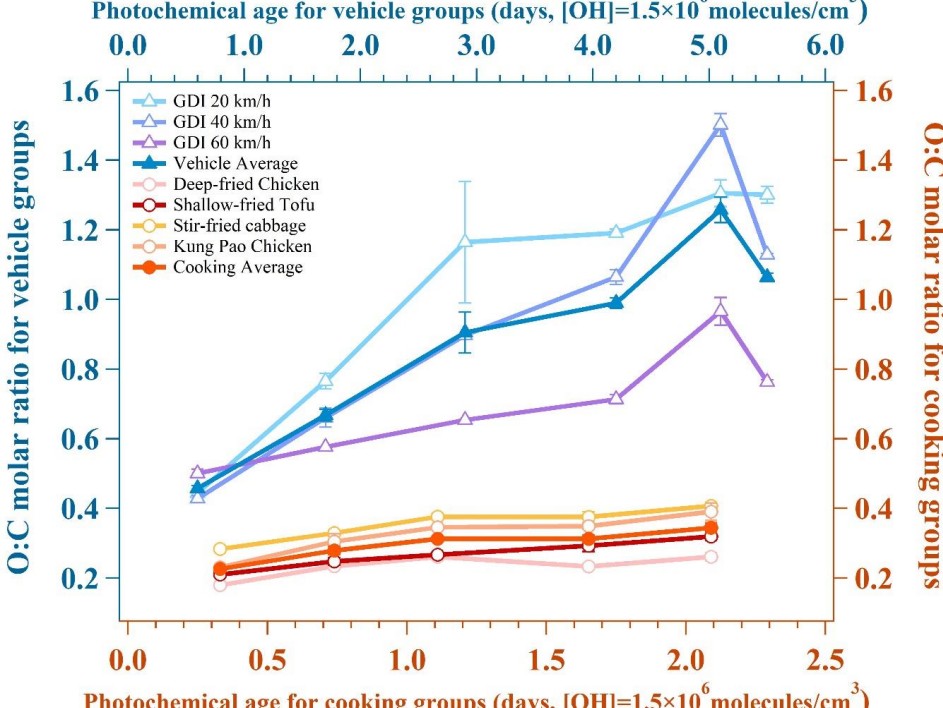


**Figure 3.** Evolution of O:C molar ratio for two urban lifestyle SOA.





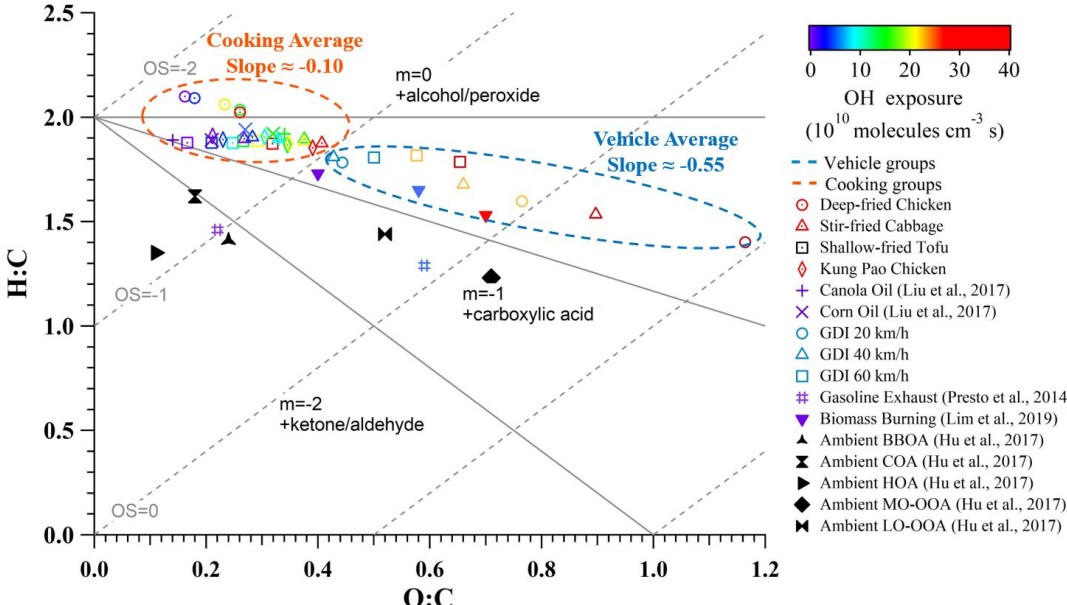

**Figure 4.** Van Krevelen diagram of OA from various sources.

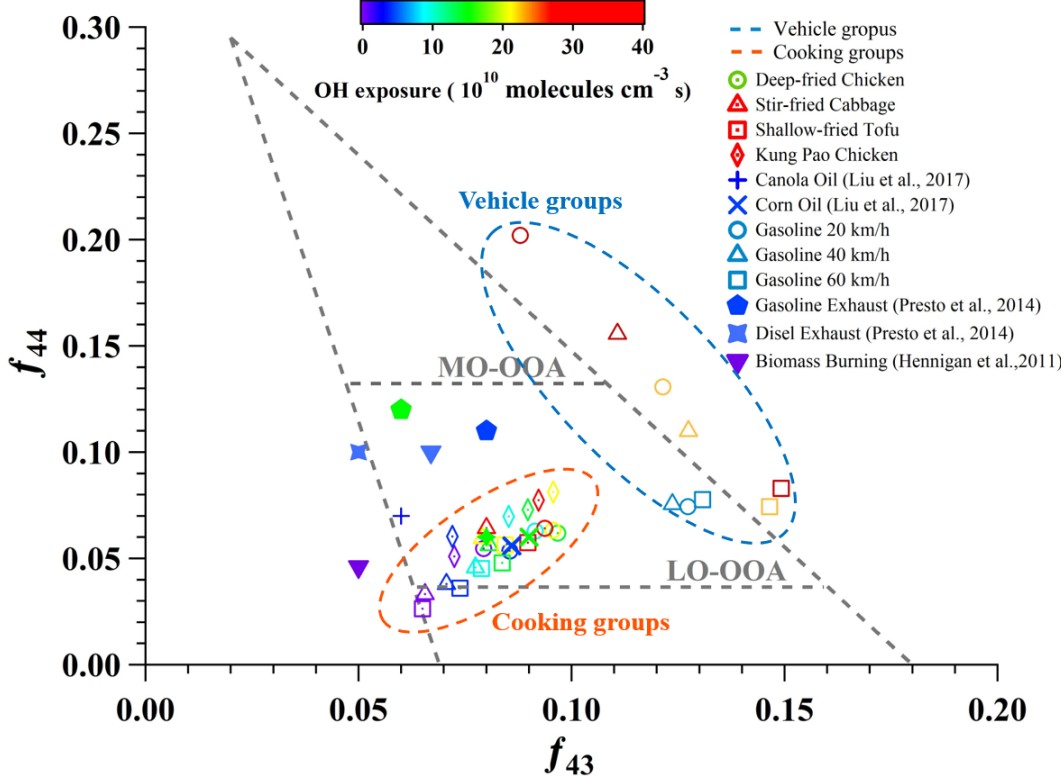

**Figure 5.** Fractions of entire organic signals at m/z 43 ($f_{43}$) vs. m/z 44 ($f_{44}$) from various sources as well as Ng triangle plot.







**Figure 6.** Average mass spectra of OA from two urban lifestyle sources.



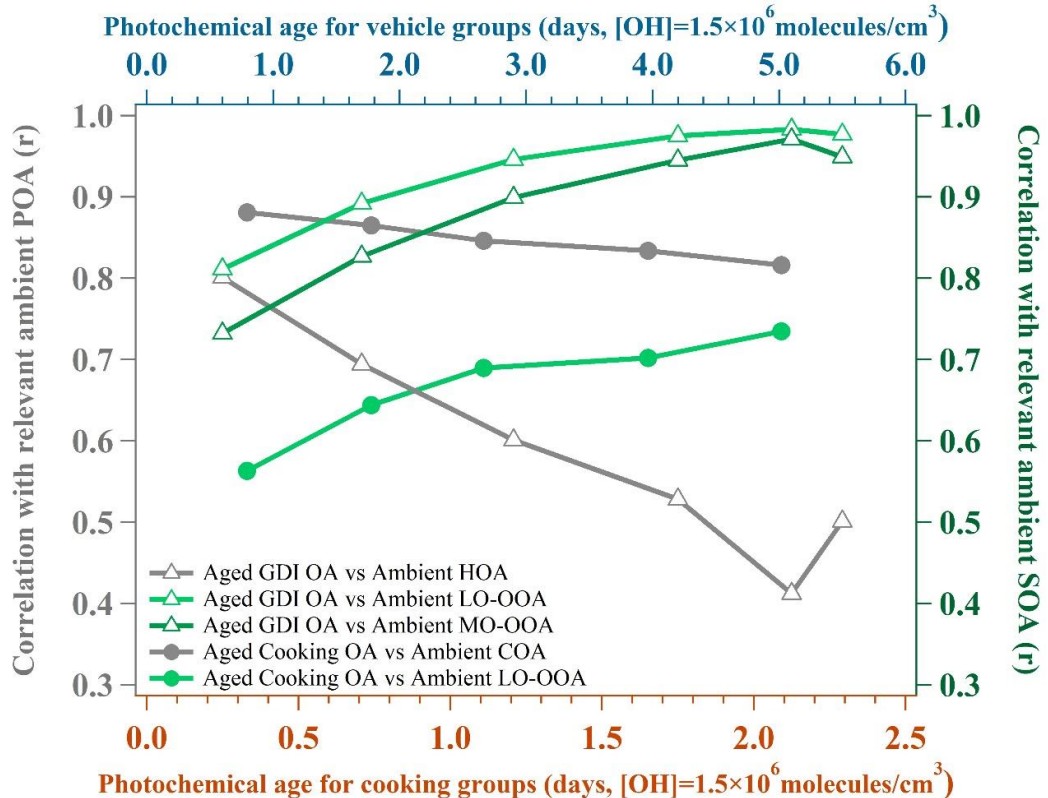

**Figure 7.** Correlation coefficients (pearson r) between the lab aged OA and published ambient PMF-OA factors with growing photochemical ages. Ambient PMF-OA factors are the average results from two field studies in Beijing (Measured at a typical urban site during autumn and winter; Autumn: Oct. 1st, 2018 – Nov. 15th, 2018; Winter: Jan. 5th, 2019 – Jan. 31st, 2019).