# Peer review of "Formation and Evolution of Secondary Organic Aerosol Derived from Urban Lifestyle Sources"

_Atmospheric Chemistry and Physics, 2021_

## Author Comment (AC1)

**I.     General Comments from Reviewer 1**

*This article characterizes secondary aerosol formation potential from vehicles and cooking that are two very important aerosol sources in urban areas. The secondary aerosol formation potential from these sources remains poorly characterized, so the topic is timely and important. Experiments done in laboratory and field seem to be comprehensive and novel in many aspects. The article is nicely written and relatively easy to follow, however it would be important to revise the English language. Also, the experiments done and instruments used need to be described better. Also, the PMF results (e.g. how the number of factors was decided, how they were identified, how many factor solutions were tested and why the were not chosen etc) needs to be described better in order to reader to understand the results.*

Reply: We appreciate the constructive comments from the reviewer on this manuscript. We have answered them point to point in the following paragraphs (the texts italicized are the reviewers' comments, the texts indented are the responses, and the texts in blue are revised parts in the new manuscript or supporting information). In addition, all changes made are marked in the revised manuscript.

We used PMF to split aged cooking OA into two OA factors. Generally, there is at least one POA and one SOA (factor 1-POA; factor 2-SOA). When we chose three or more factors, we found the elemental ratios or mass spectra of additional OA factors are quite similar to factor 1 or factor 2, which means that we can't find another new OA factor. Therefore, we finally chose 2 OA factors, one for POA and another for SOA. As Figure S5-S8 shows, the SOA factor presents a larger fraction of oxygen-containing fragments (especially in m/z 28, 29, 43, 44) and higher O/C, which is different from those of POA factors.

Revised text "The maximum SOA mass growth potentials of aged cooking SOA only ranged from 1.9-3.2 implying a mixture of POA and SOA, so its mass spectra needed to be deeply resolved by PMF in order to separate the POA and SOA (precisely, a kind of LO-OOA). Generally, there is at least one POA and one SOA (factor 1-POA; factor 2-SOA). When three or more factors were set, it was found that elemental ratios or mass spectra of additional OA factors are quite similar to factor 1 or factor 2, which means that two OA factor was the best choice (one for POA and another for SOA). As Figure S5-S8 shows, the SOA factors present a larger fraction of oxygen-containing fragments (especially in m/z 28, 29, 43, 44) and higher O/C, which is significantly different from those of POA factor."

**II. Detailed comments from Reviewer 1**

1. *L20-21: is the source of SOA the traffic and cooking or city dwellers? please clarify sentence*

   The source of SOA is the traffic and cooking. Traffic and cooking are closed to the daily life of city dwellers. We have revised this ambiguous sentence.

   > Revised text: *"Vehicle exhaust and cooking emission are closely related to the daily life of city dwellers.* Here, we defined the secondary organic aerosol (SOA) derived from vehicle exhaust and cooking emission as "Urban Lifestyle SOA", and simulated their formation using a Gothenburg potential aerosol mass reactor (Go: PAM). *"*

2. *L24: define POA*

   POA is the primary organic aerosol.

   > Revised text: *"The SOA/POA (primary organic aerosol) mass ratios……"*

3. *L24: what instrument was used to measure SOA/POA in this case? AMS or SMPS?*

   It was SMPS.

   > Revised text: *"The SOA/POA (primary organic aerosol) mass ratios of vehicle groups (107) were 44 times larger than those of cooking groups (2.38) at about 2 days of equivalent photochemical age, according to the measurement of scanning mobility particle sizer (SMPS).*"

4. *L26: define "vehicle group" and "cooking group"*

   We have added the definition in the revised manuscript.

   > Revised text: *"The vehicle exhaust and cooking emission were separately simulated, and their samples were defined as "vehicle group" and "cooking group", respectively."*

5. *L35: replace could with can.*

   Thanks for the correction.

   > Revised text: *"The OA can be divided into the primary organic aerosol (POA) and the secondary organic aerosol (SOA)."*

6. **L36: I propose reformulating: POA is directly emitted into ambient air through several sources such as coal combustion, biomass burning, vehicle exhaust, cooking procedure.**

We agree with the reviewer and modify our expression.

Revised text: *"There are many potential sources of POA, such as coal combustion, biomass burning, vehicle exhaust, cooking procedure and so forth."*

7. **L39: change to: but models typically fail to simulate.**

Thanks for the correction.

Revised text: *"Significant SOA formation has been observed in several urban areas, but* models typically fail *to simulate this phenomenon accurately."*

8. **L47-49: Please reformulate by changing "would -> can be".**

Thanks for the correction.

Revised text: *"they* can be *oxidized, distributed into particle phase and finally become the part of POA or SOA."*

9. **L52: I am not sure if Manchester meets the definition of a Megacity. please check.**

We have carefully checked the population of Manchester, and it is about half a million. Therefore, Manchester is not a megacity. We have removed Manchester from the new manuscript.

Revised text: *"take the megacity for example, in London, these two urban lifestyle sources contributed 50% of OA in average"*

10. **L52: replace lifestyle source with SOA source.**

**Original Text:** *"Vehicle and cooking emissions are important sources of OA in urban areas (Rogge et al., 1991;Rogge et al., 1993;Hu et al., 2015;Hallquist et al., 2016;Crippa et al., 2013;Mohr et al., 2012;Guo et al., 2013;Guo et al., 2012), take a megacity for example, in London, these two urban lifestyle sources contributed 50% of OA in average (Allan et al., 2010)."*

Thanks for the reviewer's comment. However, it is not very appropriate to use "SOA source" here, because vehicle and cooking are both POA and SOA sources. Furthermore, as for the study in London,

these two urban lifestyle sources can contribute 50% of OA when only considering POA, and it was not sure how much SOA was generated from these two sources according to that study. Therefore, it is better to use "urban lifestyle source" instead of "SOA source".

**11. L65: replace Lab with laboratory all through the manuscript**

We have replaced the lab with laboratory all through the manuscript as the reviewer suggested.

**12. L83, 102: define China V**

More description about China V was added to the revised manuscript as the following text shows:

Revised text "The commercial China Phase V gasoline was used as the fuel, which has equivalent octane number 92 level (RON 92), 10 ppm (v/v, max) sulfur, 25% (v/v, max) olefin, about 40% (v/v, max) aromatics, 2 mg/L Mn and no oxygenates." (Yinhui et al., 2016).

Yinhui, W., Rong, Z., Yanhong, Q., Jianfei, P., Mengren, L., Jianrong, L., Yusheng, W., Min, H., and Shijin, S.: The impact of fuel compositions on the particulate emissions of direct injection gasoline engine, Fuel, 166, 543-552, 10.1016/j.fuel.2015.11.019, 2016.

**13. L102: Is this vehicle or engine that you are using to produce the exhaust? describe the engine/vehicle (manufacturer, model, engine size, aftertreatment, mileage, year, fuel, lubricant oil) in detail. All these have a major impact to SOA formation. Also, if engine, maybe add information that this is not exactly same as vehicle, assuming there is no catalysts (like nowadays almost all have). Please add discussion about the differences between engine produced SOA and vehicle produced SOA.**

It is the Gasoline direct injection engine instead of a vehicle. The detailed information of test GDI engine equipped with catalyst system has been added in Table S2-S3.

Revised text: "More information about GDI engine can be found in Table S2-S3."

**Table S2.** Test engine information.

| Specification | GDI |
|---|---|
| Displaced Volume | 998 cc |
| Stoke | 78.6 mm |
| Bore | 73.4 mm |
| Compression ratio | 9.6 |
| Max power / engine speed | 100 kW / 6000 rpm |
| Max torque / engine speed | 205 N·m / 2000-3000 rpm |

**Table S3.** Catalyst system information.

| Specification | Three-way catalyst system |
|---|---|
| Volume | 1.19 L |
| Material | Cordierite |
| Diameter | 132.1 mm |
| Length | 87.1 mm |
| Cell | 900 /inch$^2$ |

There is a three-way catalyst system after our GDI engine. Our engine condition is indeed different from those of vehicle experiments. The main difference is that engine experiments can hold fixed speed and torque conditions, while vehicle experiments often focus on run cycles with dynamic speed and torque conditions. Unfortunately, there are few studies that directly used the GDI engine instead of the vehicle to study the SOA formation in detail, so we just compared our results with those from vehicle experiments equipped with gasoline or diesel engine.

**14. L102. Accronym GDI should be Gasoline direct injection**

We have revised it to "Gasoline direct injection".

Revised text: *"The vehicle exhaust was emitted from a Gasoline direct engine (GDI)……"*

**15. L104: please, describe the used sampling and dilution setup in detail. How the dilution air was cleaned? dirty dilution air can be major source of SOA. Was the vehicle exhaust taken directly from tailpipe? and cooking fumes from the room air? if so, what it the influence of this extra dilution/aging in the room air in the case of cooking fumes? For the cooking, maybe explain how the boiling water acts as a blank. Was a blank/zero measurements done for the vehicle? Also, I find it hard to find how these measurement points were taken. According to table there was 6 points with different OH concentrations, and 3-5 repetitions for each. Please clarify how long each point was, were the results of repetitions averaged? if so, maybe give standard deviations to values in figures/tables to describe the variability between repetitions.**

Thank the reviewer for reminding us to supply necessary details. The dilution air was ambient air (clean period), which was firstly filtered by a particle filter system (including a dryer, a filter, and an ultrafilter, SMC Inc.) to remove the particles and water. Then the dilution air was filtered by an activated carbon adsorption device to remove the VOCs. As the results of dilution air groups in Table S1, the dilution air just made a minor influence on the SOA concentration.

Vehicle exhaust from the tailpipe was first diluted by a gradient heated dilution system (6 fold) and then diluted by an unheated dilution system (5 fold). The temperature of sample flow was near indoor temperature after secondary dilution systems.

The cooking fumes were collected through the kitchen ventilator. The boiled water can be a background sample influenced by indoor air, iron wok, and ventilator.

A blank/zero measurement of vehicle experiment have also been done as following Table S1 shows. Among different OH exposure, we have done 3-5 repetitions, the results showed in the figure are their average values. Every repetition point took 2 min, which was equal to the time resolution of AMS and SMPS. The deviations have been shown in figure 2-3 in the form of an error bar. We have noted these details in figure captions.

The detailed formation has been added in Section S1, Table S1, and Figure 2-3 captions:

> Revised text "The dilution air was ambient air (clean period), which was firstly filtered by a particle filter system (including a dryer, a filter and an ultrafilter, SMC Inc.) in order to remove the particles and water. Then the dilution air was filtered by an activated carbon adsorption device, in order to remove the VOCs. Vehicle exhaust from tailpipe was first diluted by a gradient heated dilution system (6 fold) and then diluted by a unheated dilution system (5 fold). The temperature of sample flow was near indoor temperature after secondary dilution systems. The cooking fumes was collected through the kitchen ventilator. The boiled water can be a background sample influenced by indoor air, iron wok and ventilator. As the results of blank groups in Table S1, the dilution air and background interference just made minor influence to the SOA concentration."

**Table S1.** Comparison of results between blank and experimental groups (Dilution air and boiled water are two kinds of blank groups. The others are experimental groups.).

| Experiment | OH Exposure ($\times 10^{10}$ molecules·cm$^{-3}$·s) | OA Concentration (µg/m3) | Standard Deviation | Relative Standard Deviation |
|---|---|---|---|---|
| Dilution Air (cooking) | 0 | - | - | - |
| | 9.6 | 0.37 | 0.04 | 12% |
| Boiled Water | 0 | 0.04 | 0.02 | 44% |
| | 9.6 | 0.36 | 0.12 | 32% |
| Deep-fried Chicken | 0 | 12.30 | 0.49 | 4% |
| | 9.6 | 28.29 | 2.55 | 9% |
| Shallow-fried Tofu | 0 | 13.56 | 0.68 | 5% |
| | 9.6 | 21.70 | 1.08 | 5% |
| Stir-fried Cabbage | 0 | 10.75 | 0.65 | 6% |
| | 9.6 | 18.38 | 1.65 | 9% |
| Kung Pao Chicken | 0 | 6.47 | 0.52 | 8% |
| | 9.6 | 11.39 | 1.25 | 11% |
| Dilution Air (vehicle) | 0 | - | - | - |
| | 7.8 | 0.52 | 0.07 | 13% |
| GDI 20 km/h | 0 | 0.40 | 0.01 | 3% |
| | 7.8 | 19.68 | 1.48 | 8% |
| GDI 40 km/h | 0 | 0.41 | 0.01 | 3% |
| | 7.8 | 15.24 | 0.62 | 4% |
| GDI 60 km/h | 0 | 0.42 | 0.02 | 5% |
| | 7.8 | 23.23 | 4.00 | 17% |

**Figure 2.** Secondary mass growth potentials for two urban lifestyle SOA. The mass growth potentials were represented by SOA/POA mass ratios. The SMPS-1 determined the mass concentration of POA, while the SMPS-2 determined the mass concentration of aged OA, and their mass difference could be regarded as the SOA. The average data and standard deviation bars are shown in the figure.

**Figure 3.** Evolution of O:C molar ratio for two urban lifestyle OA. The O:C molar ratios are determined by HR-Tof-AMS. The average data and standard deviation bars at each gradient are shown in the figure.

16. *L117: This chapter is quite unclear and missing quite many details. Please, explain the AMS and GoPAM in more detail, e.g. in separate chapters and the rest shortly like it is now.*

The details of HR-Tof-AMS have been added to section 2.2 as follows:

Revised text "The chemical compositions of OA were measured by a high-resolution time-of-flight aerosol mass spectrometer (HR-ToF-AMS, Aerodyne Research Inc.), in which the non-refractory particles including organics, sulfate, nitrate, ammonium, and chloride were instantly vaporized by a 600°C tungsten. Next, the vaporized compounds were ionized by an electron impact (EI) ionization with 70 ev. Finally, the fragment ions were pulsed to a time-of-flight MS chamber and detected by the multi-channel plate detector (MCP) (Nash et al., 2006;DeCarlo et al., 2006). In this study, its time resolution was 2 min (precisely, 1 min for a mass sensitive V-mode, and 1 min for a high mass resolution W-mode)."

We have also supplied other parameters of HR-Tof-AMS in section 2.3.1:

Revised text "A default value (1.4) of relative ionization efficiency (RIE) of OA was adopted. Another synchronous SMPS-2 was used to correct the collection efficiency (CE) of HR-ToF-AMS by comparing their mass concentrations (Gordon et al., 2014). Before the formal experiment, the IE and RIESO4 were calculated by the comparison of HR-Tof-AMS and SMPS, when the sampling flow was generated by 300 nm ammonium nitrate and 300 nm ammonium sulfate, respectively, with an Aerosol generator (DMT Inc.). The CE was a fluctuant value influenced by the emission condition, so it was estimated by the comparison of HR-Tof-AMS (sampling after Go: PAM) and SMPS-2 (sampling after Go: PAM) during the formal experiment. The CE and $RIE_{Org}$ were theoretically different in every emission or oxidation condition, so we directly use the SMPS measurements to determine the aged OA mass concentration. As for the cooking experiment, the IE value was $7.77 \times 10^{-8}$, the RIESO4 was 1.4, the $RIE_{Org}$ was 1.4 (default value, the fluctuation of $RIE_{Org}$ was included in CE), the average CE was about 0.55 (ranged from 0.3 to 0.7). As for the vehicle experiment, the IE value was $7.69 \times 10^{-8}$, the RIESO4 was 1.3, the $RIE_{Org}$ was 1.4 (default value, the fluctuation of $RIE_{Org}$ was included in CE), the average CE was about 0.6 (ranged from 0.4 to 0.7).

The details of Go: PAM have been added to the manuscript and SI as follows:

Revised text "The internal structure of Go: PAM can be found in Figure S1."

Revised text "As Figure S1 shows, the flow reactor of Go: PAM is made of quartz glass (1) (Raesh GmbH RQ 200), which is 100 cm long and 9.6 cm in diameter. About 84 cm of the flow reactor may be illuminated by either one or two Philips TUV 30 W fluorescent tubes (2), each radiating about 10 W at 254 nm. It is enclosed in a compartment of aluminum mirrors to reduce the inhomogeneity of the photon field inside the reactor. The fluorescent tubes and quartz tubes are surrounded by a parabolic trough mirror (3), 90 deg. flat mirror (4) and 45-90 deg. flat mirrors (5). The shell of Go: PAM is composed of a sheath metal cover (6) and square tubing support structure (7)."

[Figure]

**Figure S1**. Profile of Go: PAM. (1) 9.6 cm quartz tube (2) fluorescent tube (3) parabolic trough mirror (4)90 deg. flat mirror (5) 45-90 deg. flat mirror (6) sheath metal cover (7) Square tubing support structure

17. **L120: I think the reference describing Aerodyne instrument should be: DeCarlo, Peter F., Joel R. Kimmel, Achim Trimborn, Megan J. Northway, John T. Jayne, Allison C. Aiken, Marc Gonin, et al. "Field-Deployable, High-Resolution, Time-of-Flight Aerosol Mass Spectrometer." Analytical Chemistry 78, no. 24 (December 2006): 8281–89. https://doi.org/10.1021/ac061249n.**

Thanks for the suggestion. We have added this reference to the related section.

Revised text: *"A high-resolution time-of-flight aerosol mass spectrometer (HR-ToF-AMS, Aerodyne Research Inc.) was used to identify the chemical compositions of OA (DeCarlo et al., 2006; Nash, Baer, & Johnston, 2006)."*

18. **L128: I think this should state that the measured CO2 concentrations (Model 410i, Thermo Electron Corp.) were used to conduct CO2 correction for AMS data in order to reduce the CO2 interference to organic fragments in mass spectra of HR-ToF-AMS.**

We agree with the correction.

Revised text: *"The measured $CO_2$ concentrations (Model 410i, Thermo Electron Corp.) were used to conduct $CO_2$ correction for AMS data in order to reduce the $CO_2$ interference to organic fragments in mass spectra of HR-ToF-AMS."*

19. **L125-126: You are saying that "The SMPS-1 determined the mass concentration of POA, while the SMPS-2 determined the mass concentration of aged OA, and their mass difference could be regarded as the SOA.". are these results shown in somewhere?**

These data are shown in figure 2 in the form of SOA/POA. The vehicle exhaust and cooking fume were partly collected from the tailpipe and kitchen ventilator, which means we didn't focus on the total mass of source emission. We focused more on the relative properties and parameters, like SOA/POA, elemental ratios, and mass spectra which are independent of the total mass of OA.

20. **L143: The ionization efficiency (IE), relative ionization efficiency (RIE) and collection efficiency (CE) were determined individually before data processing. How were these determined? please give the results.**

Before the formal experiment, the IE and $RIE_{SO4}$ were calculated by the comparison of HR-Tof-AMS and SMPS, when the sampling flow was generated by 300 nm ammonium nitrate and 300 nm ammonium sulfate, respectively, with an Aerosol generator (DMT Inc.). The CE was a fluctuant value influenced by the emission composition, so it was estimated by the comparison of HR-Tof-AMS (sampling after Go: PAM) and SMPS-2 (sampling after Go: PAM) during the formal experiment. The CE and $RIE_{Org}$ were theoretically different in every emission or oxidation condition, so we directly use the SMPS measurements to determine the aged OA mass concentration.

As for the cooking experiment, the IE value was $7.77 \times 10^{-8}$, the $RIE_{SO4}$ was 1.4, the $RIE_{Org}$ was 1.4 (default value, the fluctuation of $RIE_{Org}$ was included in CE), the average CE was about 0.55 (ranged from 0.3 to 0.7).

As for the vehicle experiment, the IE value was $7.69 \times 10^{-8}$, the $RIE_{SO4}$ was 1.3, the $RIE_{Org}$ was 1.4 (default value, the fluctuation of $RIE_{Org}$ was included in CE), the average CE was about 0.6 (ranged from 0.4 to 0.7).

We have added a new part in section 2.3.1 as follows:

> Revised text: "Before the formal experiment, the IE and $RIE_{SO4}$ were calculated by the comparation of HR-Tof-AMS and SMPS, when the sampling flow were generated by 300 nm ammonium nitrate and 300 nm ammonium sulfate, respectively, with an Aerosol generator (DMT Inc.). The CE was a fluctuant value influenced by the emission composition, so it was estimated by the comparation of HR-Tof-AMS (sampling after Go: PAM) and SMPS-2 (sampling after Go: PAM) during the formal experiment. The CE and $RIE_{Org}$ were theoretically different in every emission or oxidation condition, so we directly use the SMPS measurements to determine the aged OA mass concentration. As for the cooking experiment, the IE value was $7.77 \times 10^{-8}$, the $RIE_{SO4}$ was 1.4, the $RIE_{Org}$ was 1.4 (default value, the fluctuation of $RIE_{Org}$ was included in CE), the average CE was about 0.55 (ranged from 0.3 to 0.7). As for the vehicle experiment, the IE value was $7.69 \times 10^{-8}$, the $RIE_{SO4}$ was 1.3, the $RIE_{Org}$ was 1.4 (default value, the fluctuation of $RIE_{Org}$ was included in CE), the average CE was about 0.6 (ranged from 0.4 to 0.7)."

*21. L146-147: please give the range of CE values and average CE value*

As for cooking experiment, the IE value was $7.77 \times 10^{-8}$, the $RIE_{SO4}$ was 1.4, the average CE was about 0.55 (ranged from 0.3 to 0.7).

As for vehicle experiment, the IE value was $7.69 \times 10^{-8}$, the $RIE_{SO4}$ was 1.3, the average CE was about 0.6 (ranged from 0.4 to 0.7).

We have added new part in section 2.3.1 as follows:

Revised text: "Before the formal experiment, the IE and $RIE_{SO4}$ were calculated by the comparation of HR-Tof-AMS and SMPS, when the sampling flow were generated by 300 nm ammonium nitrate and 300 nm ammonium sulfate, respectively, with an Aerosol generator (DMT Inc.). The CE was a fluctuant value influenced by the emission composition, so it was estimated by the comparation of HR-Tof-AMS (sampling after G₀: PAM) and SMPS-2 (sampling after G₀: PAM) during the formal experiment. The CE and $RIE_{Org}$ were theoretically different in every emission or oxidation condition, so we directly use the SMPS measurements to determine the aged OA mass concentration. As for cooking experiment, the IE value was $7.77 \times 10^{-8}$, the $RIE_{SO4}$ was 1.4, the $RIE_{Org}$ was 1.4 (default value, the fluctuation of $RIE_{Org}$ was included in CE), the average CE was about 0.55 (ranged from 0.3 to 0.7). As for vehicle experiment, the IE value was $7.69 \times 10^{-8}$, the $RIE_{SO4}$ was 1.3, the $RIE_{Org}$ was 1.4 (default value, the fluctuation of $RIE_{Org}$ was included in CE), the average CE was about 0.6 (ranged from 0.4 to 0.7)."

**22. L147: SOA was defined earlier as SMPS2-SMPS1. How does this results gained from PMF compare to SMPS results?**

The SMPS directly measured the concentration of POA and SOA, and PMF separated the POA and SOA by mathematical method. It is interesting to compare their results. We have been added this part in Table S4. As for deep-fried chicken and stir-fried cabbage, two methods tend to obtain similar results. As for relative lower photochemical age (less than 1.1 days), two methods tend to obtain the similar results. When it comes to the mass ratio, the direct measurement by two sets of SMPS seem to be more accurate than the estimation by the mathematical PMF.

Table S4. The comparison of SOA/POA between SMPS and AMS-PMF results. "SOA/POA (SMPS)" means the mass ratio gained from SMPS-1 and SMPS-2. "SOA/POA (AMS-PMF)" means the mass ratio gained from PMF analysis of aged OA measured by HR-Tof-AMS.

| Photochemical Age (days, [OH]=$1.5 \times 10^6$ molecules·cm$^{-3}$) | Deep Fried Chicken | | Shallow-fried Tofu | | Stir-fried cabbage | | Kung Pao Chicken | | Cooking Average | |
|---|---|---|---|---|---|---|---|---|---|---|
| | SOA/POA (AMS-PMF) | SOA/POA (SMPS) | SOA/POA (AMS-PMF) | SOA/POA (SMPS) | SOA/POA (AMS-PMF) | SOA/POA (SMPS) | SOA/POA (AMS-PMF) | SOA/POA (SMPS) | SOA/POA (AMS-PMF) | SOA/POA (SMPS) |
| 0.3 | 0.63 | 0.46 | 0.34 | 0.34 | 0.50 | 0.41 | 0.53 | 0.51 | 0.50 | 0.43 |
| 0.7 | 1.84 | 1.29 | 1.29 | 0.61 | 0.93 | 0.71 | 0.87 | 0.77 | 1.23 | 0.84 |
| 1.1 | 2.21 | 2.13 | 1.97 | 0.81 | 1.87 | 1.14 | 1.44 | 1.22 | 1.87 | 1.33 |
| 1.7 | 2.30 | 2.41 | 3.32 | 1.27 | 1.95 | 1.57 | 4.57 | 1.92 | 3.03 | 1.79 |
| 2.1 | 3.23 | 3.16 | 4.50 | 1.81 | 2.04 | 2.05 | 6.28 | 2.48 | 4.01 | 2.38 |

**23. L177: I don't really understand what this means. Please clarify the sentence: "The mixing and wall loss conditions have already met our experiment needs."**

We are sorry that this is a meaningless sentence. We have removed this sentence now. The detailed evaluation of Go: PAM could be found in Figure S4.

**24. L183: should this be secondary aerosol formation potential?**

Yes, it is the secondary aerosol formation potential. The previous expression is a bit unclear. The revised expressions are as follows:

Revised text: *"3.1 Secondary Formation Potential of the Urban Lifestyle SOA."*

Revised text: *"As Figure 2 shows, the mass growth potentials of two urban lifestyle SOA were quite different.* The mass growth potentials were represented by SOA/POA mass ratios.*"*

**25. L184: How do you know the functionalization increased as the photochemical age increased?**

*Original text "Their SOA/POA mass ratios both increased gradually through functionalization reactions and finally reached the peak after 2-3 days of equivalent photochemical age".*

Thanks for the careful reading. We didn't emphasize "*functionalization increased as the photochemical age increased*". We wanted to state that SOA/POA mass ratios increased gradually and finally **reached the peak** after 2-3 days of equivalent photochemical age. In this process, SOA may form through functionalization reactions. We'd like to remove the word "functionalization" to avoid misunderstanding.

Revised text: "Their SOA/POA mass ratios both increased gradually and finally reached the peak after 2-3 days of equivalent photochemical age…..."

**26. L184-185: How the SOA is defined here? is this all mass (POA+SOA) after the go:PAM or is it the SOA separated by PMF or calculated from the SMPS? please, define the terms you use and use them systematically. Include the same information to the figure captions.**

Thanks for the advice. Here, SOA is the SMPS-2 (aged OA)-SMPS-1(POA). We have defined the measured SOA in section 2.2. In section 3.1, we need to define it again to make the article easy to follow. The additional definition is as follows:

Revised text: "The SMPS-1 determined the mass concentration of POA, while the SMPS-2 determined the mass concentration of aged OA, and their mass difference could be regarded as the SOA."

**27. L186: define term "mass growth potential"**

The additional explanation of this term is as follows:

Revised text: "The mass growth potentials were represented by SOA/POA mass ratios. The SMPS-1 determined the mass concentration of POA, while the SMPS-2 determined the mass concentration of aged OA, and their mass difference could be regarded as the SOA."

**28. L191-194: are these aromatics, cycloalkanes, fatty acids etc compounds you measured or found from literature? if literature, please formulate the sentences so that it is clear that this is information found from literature.**

These compounds are found from other works. We didn't provide the direct measurements of gaseous precursors, we have removed these further speculations to avoid misunderstanding.

**29. L219: please define f43 and f44**

$f43$ and $f44$ are the signal fraction of organic fragments. We have defined them in the new manuscript:

Revised text: "As shown in Figure 5, the signal fraction of organic fragments at m/z 43 ($f43$) and m/z 44 ($f44$) has been widely adopted to represent the oxidation process of OA."

**30. L222: remove apparently**

As the reviewer suggested, the word "apparently" was removed.

Revised text: "The datasets of vehicle and cooking groups fell along in different regions and showed different variations in the plot."

**31. L247-249: are the fx fractions average fractions for all measurement points with different OH-exposure and different speeds? maybe give range or standard deviation to describe the variation within the dataset.**

Yes, it is the average fractions, and the standard deviations have been added as the following text shows:

Revised text "For average vehicle LO-OOA mass spectra, the prominent peaks were m/z 43 ($f_{43}$=0.133±0.003), 44 ($f_{44}$=0.077±0.001), 29 ($f_{29}$=0.076±0.003), 28 ($f_{28}$=0.066±0.001), 41 ($f_{41}$=0.051±0.005), 55 ($f_{55}$=0.043± 0.004) dominated by $C_2H_3O^+$, $C_3H_7^+$, $CO_2^+$, $CHO^+$, $C_2H_5^+$, $CO^+$, $C_3H_5^+$, $C_3H_3O^+$ and $C_4H_7^+$ respectively, while the prominent peaks of average vehicle MO-OOA were m/z 44 ($f_{44}$=0.146±0.060), 28 ($f_{28}$=0.134±0.062), 43 ($f_{43}$=0.117±0.033), 29 ($f_{29}$=0.071±0.014), 45 ($f_{45}$=0.032±0.007), 27 ($f_{27}$=0.030±0.009) dominated by $CO_2^+$, $CO^+$, $C_2H_3O^+$, $CHO^+$, $C_2H_5^+$, $CHO_2^+$, $C_2H_5O^+$ and $C_2H_3^+$ respectively."

Revised text:"For average cooking LO-OOA, it was less oxidized than those from vehicle groups, whose prominent peaks were m/z 43 ($f_{43}$=0.097±0.008), 44 ($f_{44}$=0.065±0.010), 29 ($f_{29}$=0.065±0.013), 41 ($f_{41}$=0.058± 0.008), 55 ($f_{55}$=0.056±0.006), 28 ($f_{28}$=0.053±0.011) dominated by $C_2H_3O^+$, $C_3H_7^+$, $CO_2^+$, $CHO^+$, $C_2H_5^+$, $C_3H_5^+$, $C_3H_3O^+$, $C_4H_7^+$ and $CO^+$ respectively."

*32. Figure/table captions: please, add all necessary information to figure/table caption. E.g. is the shown data average values, are the shown bars standard deviations, which instrument was used to measure data, etc.*

We have modified the caption of Figure 2-7 as the reviewer suggested.

Revised text:

**Figure 2.** Secondary mass growth potentials for two urban lifestyle SOA. The mass growth potentials were represented by SOA/POA mass ratios. The SMPS-1 determined the mass concentration of POA, while the SMPS-2 determined the mass concentration of aged OA, and their mass difference could be regarded as the SOA. The average data and standard deviation bars are shown in the figure.

**Figure 3.** Evolution of O:C molar ratio for two urban lifestyle OA. The O:C molar ratios are determined by HR-Tof-AMS. The average data and standard deviation bars at each gradient are shown in the figure.

**Figure 4.** Van Krevelen diagram of OA from various sources. The O:C and H:C are determined by HR-Tof-AMS. The average data at each gradient are shown in the figure.

**Figure 5.** Fractions of entire organic signals at m/z 43 ($f_{43}$) vs. m/z 44 ($f_{44}$) from various sources as well as Ng triangle plot. The $f_{43}$ and $f_{44}$ are determined by HR-Tof-AMS. The average data at each gradient are shown in the figure.

**Figure 6.** Average mass spectra of OA from two urban lifestyle sources. The numbered symbols represent the m/z values with relatively large fractions. The gray symbols represent the fragments that mainly come from

hydrocarbon-like fragments and the green symbols represent those mainly come from oxygen-containing fragments. The mass spectra are determined by HR-Tof-AMS. The average data are shown in the figure.

**Figure 7.** Correlation coefficients (Pearson r) between the laboratory aged OA and published ambient PMF-OA factors with growing photochemical ages. Ambient PMF-OA factors are the average results from two field studies in Beijing (Measured at a typical urban site during autumn and winter; Autumn: Oct. 1st, 2018 – Nov. 15th, 2018; Winter: Jan. 5th, 2019 – Jan. 31st, 2019). The unit mass resolution mass spectra are determined by HR-Tof-AMS.

*33. Figure 2. Define what data was used for SOA/POA in the figure.*

It is the SMPS data. We have modified the caption of figure 2.

Revised text: **"Figure 2.** Secondary mass growth potentials for two urban lifestyle SOA. The mass growth potentials were represented by SOA/POA mass ratios. The SMPS-1 determined the mass concentration of POA, while the SMPS-2 determined the mass concentration of aged OA, and their mass difference could be regarded as the SOA. The average data and standard deviation bars are shown in the figure."

---

## Author Comment (AC2)

**I. General Comments from Reviewer 2**

*Zhang et al. measured SOA formation from emissions from a GDI engine and four types of food cooking emissions that were exposed to OH radicals in a Go:PAM. Equivalent atmospheric aging timescales of up to 5 days were studied. The SOA/POA ratio was approximately 100 for the oxidized vehicle exhaust and 2 for the oxidized cooking exhaust. Higher SOA oxidation states were observed for the oxidized vehicle exhaust. AMS spectra of the oxidized emissions were examined and compared to ambient LO-OOA, MO-OOA, and COA factors resolved using PMF. The studies are well motivated. In its current form, there are too many important experimental details that are missing for me to support publication in ACP.*

Reply: We appreciate the constructive comments from the reviewer on this manuscript. We have added plenty of parts to method and discussion to enrich the whole article. In the revised manuscript, we have almost rewritten the SI and reprocessed the data of cooking POA and SOA by a new method. Besides, we have answered the reviewer's questions point to point in the following paragraphs (the texts italicized are the comments, the texts indented are the responses, and the texts in blue are revised parts in the new manuscript). In addition, all changes made are marked in the revised manuscript.

**II. Detailed comments from Reviewer 2**

*1.Critical details about the sampling inlets between the source emissions and the OFR (e.g. tubing length, diameter, material, residence time) that could influence the penetration efficiency and/or delays in transmission to the OFR (e.g. Pagonis et al, 2017) are missing. If these conditions vary between vehicular exhaust and cooking exhaust measurements, that is one of several potentially important variables that could complicate direct comparison of results between the two studies.*

Thank the reviewer for reminding us to supply necessary details. The sample inlets between the source emissions and the OFR mainly contained a dilutor, sampling lines ,and silicon tubes. Both vehicle and cooking fumes were diluted at a constant ratio by a Dekati Dilutor (e-Diluter, Dekati Ltd.). The dilution air was ambient air (clean period), which was firstly filtered by a particle filter system (including a dryer, a filter ,and an ultrafilter, SMC Inc.) to remove the particles and water. Then the dilution air was filtered by an activated carbon adsorption device to remove the VOCs. Vehicle exhaust from the tailpipe was first diluted by a gradient heated dilution system (6 fold) and then diluted by an unheated dilution system (5 fold). The temperature of sample flow was near indoor temperature after secondary dilution systems. The cooking fumes were collected through the kitchen ventilator. The boiled water can be a

background sample influenced by indoor air, iron wok ,and ventilator. Besides, a temperature controller and heat insulation cotton were wrapped around the sampling pipelines to prevent freshly warm gas from condensing on the pipe wall. The sampling lines were composed of stainless steel and black carbon tubes. Silicon tubes were used to dry the emissions before they entered measuring instruments. As the results of blank groups in Table S1, the dilution air and background interference just made a minor influence on the SOA concentration. We have added details in SI and Table S1.

Revised text "The dilution air was ambient air (clean period), which was firstly filtered by a particle filter system (including a dryer, a filter and an ultrafilter, SMC Inc.) in order to remove the particles and water. Then the dilution air was filtered by an activated carbon adsorption device, in order to remove the VOCs. Vehicle exhaust from tailpipe was first diluted by a gradient heated dilution system (6 fold) and then diluted by a unheated dilution system (5 fold). The temperature of sample flow was near indoor temperature after secondary dilution systems. The cooking fumes was collected through the kitchen ventilator. The boiled water can be a background sample influenced by indoor air, iron wok and ventilator. As the results of blank groups in Table S1, the dilution air and background interference just made minor influence to the SOA concentration."

**Table S1.** Comparison of results between blank and experimental groups (Dilution air and boiled water are two kinds of blank groups. The others are experimental groups.).

| Experiment | OH Exposure ($\times 10^{10}$ molecules·cm$^{-3}$·s) | OA Concentration (µg/m3) | Standard Deviation | Relative Standard Deviation |
|---|---|---|---|---|
| Dilution Air (cooking) | 0 | - | - | - |
| | 9.6 | 0.37 | 0.04 | 12% |
| Boiled Water | 0 | 0.04 | 0.02 | 44% |
| | 9.6 | 0.36 | 0.12 | 32% |
| Deep-fried Chicken | 0 | 12.30 | 0.49 | 4% |
| | 9.6 | 28.29 | 2.55 | 9% |
| Shallow-fried Tofu | 0 | 13.56 | 0.68 | 5% |
| | 9.6 | 21.70 | 1.08 | 5% |
| Stir-fried Cabbage | 0 | 10.75 | 0.65 | 6% |
| | 9.6 | 18.38 | 1.65 | 9% |
| Kung Pao Chicken | 0 | 6.47 | 0.52 | 8% |
| | 9.6 | 11.39 | 1.25 | 11% |
| Dilution Air (vehicle) | 0 | - | - | - |
| | 7.8 | 0.52 | 0.07 | 13% |
| GDI 20 km/h | 0 | 0.40 | 0.01 | 3% |
| | 7.8 | 19.68 | 1.48 | 8% |
| GDI 40 km/h | 0 | 0.41 | 0.01 | 3% |
| | 7.8 | 15.24 | 0.62 | 4% |
| GDI 60 km/h | 0 | 0.42 | 0.02 | 5% |
| | 7.8 | 23.23 | 4.00 | 17% |

The details of Go: PAM have been added to the SI as following:

As Figure S1 shows, the flow reactor of Go: PAM is made of quartz glass (1) (Raesh GmbH RQ 200), which is 100 cm long and 9.6 cm in diameter. About 84 cm of the flow reactor may be illuminated by either one or two Philips TUV 30 W fluorescent tubes (2), each radiating about 10 W at 254 nm. It is enclosed in a compartment of

aluminum mirrors to reduce the inhomogeneity of the photon field inside the reactor. The fluorescent tubes and quartz tubes are surrounded by a parabolic trough mirror (3), 90 deg. flat mirror (4) and 45-90 deg. flat mirrors (5). The shell of Go: PAM is composed of a sheath metal cover (6) and square tubing support structure (7).

[Figure]

**Figure S1**. Profile of Go: PAM. (1) 9.6 cm quartz tube (2) fluorescent tube (3) parabolic trough mirror (4)90 deg. flat mirror (5) 45-90 deg. flat mirror (6) sheath metal cover (7) Square tubing support structure

The vehicle exhaust and cooking fume were partly collected from the tailpipe and kitchen ventilator, which means we didn't focus on the total mass of source emission. We focused more on the relative properties and parameters like SOA/POA, elemental ratios and mass spectra which are independent of the total mass of OA.

*2.It is not clear to me why different OFR conditions were used in the vehicle exhaust and food cooking experiments. For example, the residence time was 110 s and the RH was 44-49% in the GDI exhaust studies, compared to 55 seconds' residence time and RH = 18-23% in the food cooking studies. This makes it more challenging to directly compare results from the different studies – for example, although there may be overlap in OH exposure between the different experiments, timescales for gas-to-particle condensation and any humidity-dependent heterogenous chemistry are different.*

Thanks for the reviewer's careful and patient review. We agree with the reviewer that it is a bit challenging to compare the results from different studies, because many different experiment conditions may make influences on the SOA formation, especially for the gas-to-particle condensation and humidity-heterogenous chemistry. The residence time is influenced by the flow rate of measurement instruments, and the humidity is influenced by the intrinsic property of the vehicle and cooking plumes. We continuously measured several times at each condition, and the standard deviations are shown in the form of error bars in Figure 1-2. The residence time may be long enough for gas-to-particle distribution during each experiment, because their error bars in Figure 1-2 are minor. Assuming that we collect three data groups during one experiment condition, if the residence time is not enough, the last group may leave residual particle or gas phase

compounds to the next group, resulting in different measurement results and big standard deviations (large error bars in figures).

Relative humidity can influence the photochemical or aqueous-phase processing of SOA, it is stated that the aerosol liquid water content may show a linear increase as a function of RH (at RH > 60%) during the three seasons in Beijing, indicating the potential impacts of aqueous-phase processing at high RH levels (Xu et al., 2017). Therefore, RH 18-23% or RH 44-49% are both relatively low, and photochemical oxidation may still play the leading role in these two experiments. In the future, we really hope we could strictly control the temperature, RH, and other conditions to deeply investigate how the aerosol ages as a function of equivalent days of atmospheric oxidation, under certain gas-phase and heterogeneous oxidations.

Xu, W., Han, T., Du, W., Wang, Q., Chen, C., Zhao, J., Zhang, Y., Li, J., Fu, P., Wang, Z., Worsnop, D. R., and Sun, Y.: Effects of Aqueous-Phase and Photochemical Processing on Secondary Organic Aerosol Formation and Evolution in Beijing, China, Environmental science & technology, 51, 762-770, 10.1021/acs.est.6b04498, 2017.

*3.More detailed information about the gas-phase measurements are necessary to interpret the results. For example, it is not clear to me whether more SOA is formed from aging of vehicle emissions because the VOC concentrations are higher, the SOA yields of those VOCs are larger, or both. Please add table(s) showing the list of VOCs that were measured from each source, their emission factors relative to $CO_2$, and their OH rate coefficients that were used to calculate the total external OH reactivity.*

Thanks for the reviewer's advice. The VOCs are measured at the inlet of Go: PAM in order to determine the OHR. We have divided them into 5 types including alkane, alkene, aromatic, O-VOCs (Oxidized VOCs, mainly included aldehyde and ketone), and X-VOCs (halogenated-VOCs) using the measurement of GC-MS (Gas Chromatography-Mass Spectrometry, GC-7890, MS-5977, Agilent Technologies Inc). The high resolution of gaseous components and their SOA or $O_3$ yields are not the main focus of this manuscript, and the detailed information of VOCs, S/I VOCs are designed to write another article, so we just list the brief result of VOCs and their $K_{OH}$ in Table S5-S6 here.

Revised text:"The detailed information of gaseous compounds and their KOH can be found in Table S5-S6. The KOH for each specie was taken from the updated Carter research results (http://www.engr.ucr.edu/~carter/reactdat.htm, last access: 24 February 2021)."

**Table S4.** VOCs measured by GC-MS at the inlet of Go: PAM.

| Expriment | TVOCs (ppbV) | Alkane (%) | Alkene (%) | Aromatic (%) | O-VOC (%) | X-VOC (%) |
|---|---|---|---|---|---|---|
| GDI 20 km/h | 33 | 60% | 6% | 12% | 13% | 9% |
| GDI 40 km/h | 35 | 55% | 7% | 13% | 13% | 12% |
| GDI 60 km/h | 29 | 54% | 6% | 12% | 14% | 13% |
| Deep-fried Chicken | 139 | 21% | 7% | 6% | 29% | 37% |
| Shallow-fried Tofu | 124 | 57% | 9% | 10% | 18% | 7% |
| Stir-fried Cabbage | 127 | 48% | 8% | 14% | 21% | 10% |
| Kung Pao Chicken | 189 | 64% | 8% | 11% | 5% | 13% |

**Table S5.** $K_{OH}$ of major species in Go: PAM.

| Species | $K_{OH}$ (cm$^{-3}$·molecules$^{-1}$·s$^{-1}$) |
|---|---|
| **Alkanes** | |
| Ethane | 2.48E-13 |
| iso-Pentane | 3.59E-12 |
| Propane | 1.09E-12 |
| n-Butane | 2.36E-12 |
| iso-Butane | 2.12E-12 |
| n-Pentane | 3.79E-12 |
| 2,3-Dimethylbutane | 5.77E-12 |
| 3-Methylpentane | 5.19E-12 |
| n-Hexane | 5.19E-12 |
| n-Butane | 2.36E-12 |
| 1,2-Dichloroethane | 2.39E-13 |
| 2,3-Dimethylpentane | 1.50E-12 |
| 3-Methylpentane | 5.19E-12 |
| Methylcyclopentane | 8.60E-12 |
| 2-Methylpentane | 5.19E-12 |
| 2-Methylheptane | 7.00E-12 |
| n-Heptane | 6.76E-12 |
| **Alkenes** | |
| Ethylene | 8.52E-12 |
| Isoprene | 1.00E-10 |
| Propene | 2.62E-11 |
| trans-2-Pentene | 6.69E-11 |
| **Aromatics** | |
| m/p-Xylene | 1.87E-11 |
| Toluene | 5.63E-12 |
| 1,2,4-Trimethylbenzene | 3.25E-11 |
| o-Xylene | 1.36E-11 |
| Benzene | 1.22E-12 |
| m/p-Xylene | 1.87E-11 |
| **O-VOCs** | |
| Acetaldehyde | 1.50E-11 |
| Acetone | 1.70E-13 |
| MTBE | 2.93E-12 |
| MethylEthylKetone | 1.22E-12 |
| MethylVinylKetone | 2.00E-11 |
| n-Hexanal | 2.99E-11 |
| Acrolein | 2.00E-11 |
| n-Pentanal | 2.79E-11 |
| **X-VOCs** | |
| Tetrachloroethylene | 1.59E-13 |
| MethyleneChloride | 1.48E-13 |
| Freon | 0.00E+00 |
| Chloroform | 1.03E-13 |
| Chloromethane | 4.30E-14 |
| **Inorganic** | |
| $SO_2$ | 9.00E-13 |
| $NO_x$ | 1.00E-11 |

*4.L96: A field study at IAP is mentioned here, but it is not clear until much later (L254) that results from this study are (I think) already published in the Li et al. (2020a) paper that is referenced much later in the manuscript.*

Thanks for the comments. The field study at IAP is used to compare its ambient OA mass spectra with our laboratory OA mass spectra. It is surely a published result in the Li et al. (2020a), and we used the original data of that published article. It may cause ambiguity, so we have removed the description in the Method section (previous L96) to Discussion section 3.4 (previous L254).

*5.L112: Given the presumably large emission factors of unsaturated fatty acids in the cooking emissions, and their corresponding fast reaction rate coefficients with ozone, it would have been useful to conduct control measurements to measure the ozonolysis products of the cooking emissions. Why were those experiments not performed here?*

We agree with the reviewer that it would be better to quantify the unsaturated fatty acids and their further influence on SOA formation. However, the AMS, SMPS, or general GC-MS can't measure the concentration of unsaturated fatty acids emitted from cooking. Unsaturated and saturated fatty acids are important components of cooking emission, but their quantifications are indeed very difficult and time-consuming. Fortunately, in this experiment, online CIMS is operated and we have collected quartz filter for Orbitrap-MS. However, these further analyses need standard materials and complicated MS analysis. We'd like to present these results in other articles in the future.

Whereas, we have done the $O_3$ oxidation groups during the cooking experiment as Table S7 shows. There is no significant increase in OA mass when we just add O3 with UV off, comparing to those of OH oxidation groups (input O3 with UV on). We admit that $O_3$ itself indeed could influence the formation of SOA, but it is hard to study this topic in this article (we mainly consider $O_3$ as the material of OH under certain water vapor and UV levels), we hope we can do more comprehensive researches in the future. Besides, a flow reactor exposure estimator was also used in this study (Peng et al., 2016). This estimator could evaluate the potential non-OH reactions in the flow reactor such as the photolysis of VOCs, the reactions with $O(^1D)$, $O(^3P)$, and $O_3$. Our results showed that non-OH reactions (including direct reaction with $O_3$) were not significant except for the photolysis of acetylacetone. But there is no acetylacetone from vehicle exhaust or cooking emission according to our measurements and previous studies. Acetylacetone was usually considered as a kind of VOCs emitted from industrial production (Ji et al., 2020). Therefore, its potential photolysis wouldn't take place during our cooking conditions, and OH

reactions still played the dominant role. The revised text in SI is as follows:

**Table S7.** Comparison of primary (no O3, UV OFF), O$_3$ oxidation (certain O$_3$, UV OFF) and OH oxidation (certain O$_3$, UV ON) results during the cooking experiment.

| Experiment | Input O$_3$ concentration (ppbV) | UV | OH Exposure (×10$^{10}$ molecules·cm$^{-3}$·s) | OA Concentration (μg/m3) | Standard Deviation | Relative Standard Deviation |
|---|---|---|---|---|---|---|
| Dilution Air (cooking) | - | OFF | 0 | - | - | - |
|  | - | ON | 9.6 | 0.37 | 0.04 | 12% |
| Boiled Water | - | OFF | 0 | 0.04 | 0.02 | 44% |
|  | - | ON | 9.6 | 0.36 | 0.12 | 32% |
| Deep-fried Chicken | - | OFF | 0 | 12.30 | 0.49 | 4% |
|  | 1183 | OFF | - | 14.50 | 0.20 | 1% |
|  | 1183 | ON | 9.6 | 28.29 | 2.55 | 9% |
| Shallow-fried Tofu | - | OFF | 0 | 13.56 | 0.68 | 5% |
|  | 1183 | OFF | - | 14.79 | 3.25 | 22% |
|  | 1183 | ON | 9.6 | 21.70 | 1.08 | 5% |
| Stir-fried Cabbage | - | OFF | 0 | 10.75 | 0.65 | 6% |
|  | 1183 | OFF | - | 12.70 | 0.72 | 6% |
|  | 1183 | ON | 9.6 | 18.38 | 1.65 | 9% |
| Kung Pao Chicken | - | OFF | 0 | 6.47 | 0.52 | 8% |
|  | 1183 | OFF | - | / | / | / |
|  | 1183 | ON | 9.6 | 11.39 | 1.25 | 11% |

Revised Text:" Except for the off-line calibration based on the decay of SO$_2$, a flow reactor exposure estimator was also used in this study (Peng et al., 2016). The OH exposures calculated by both methods showed a good correlation (Figure S1&S2). This estimator could also evaluate the potential non-OH reactions in the flow reactor such as the photolysis of VOCs, the reactions with O($^1$D), O($^3$P), and O$_3$. Our results showed that non-OH reactions were not significant except for the photolysis of acetylacetone. But there is no acetylacetone from vehicle exhaust or cooking emission according to our measurements and previous studies. Acetylacetone was usually considered as a kind of VOCs emitted from industrial production (Ji et al., 2020). Therefore, its potential photolysis wouldn't take place during our cooking conditions, and OH reactions still played the dominant role. Besides, Table S7 shows the comparison of primary (no O$_3$, UV OFF), O$_3$ oxidation (certain O$_3$, UV OFF), and OH oxidation (certain O$_3$, UV ON) results during cooking experiment. There is no significant increase in OA mass when we just add O$_3$ with UV off, comparing to those of OH oxidation groups (input O3 with UV on). Overall, our Go: PAM could reasonably simulate the OH oxidation process of cooking OA in ambient."

Peng, Z., Day, D. A., Ortega, A. M., Palm, B. B., Hu, W., Stark, H., Li, R., Tsigaridis, K., Brune, W. H., and Jimenez, J. L.: Non-OH chemistry in oxidation flow reactors for the study of atmospheric chemistry systematically examined by modeling, Atmospheric Chemistry and Physics, 16, 4283-4305, 10.5194/acp-16-4283-2016, 2016.

Ji, Y., Qin, D., Zheng, J., Shi, Q., Wang, J., Lin, Q., Chen, J., Gao, Y., Li, G., and An, T.: Mechanism of the atmospheric chemical transformation of acetylacetone and its implications in night-time second organic aerosol formation, The Science of the total environment, 720, 137610, 10.1016/j.scitotenv.2020.137610, 2020

*6.L136: The information in Tables 1-2 indicates that the sample line temperature was 20-25ºC, presumably at or close to room temperature. How did the authors determine that this temperature was sufficient to "prevent freshly warmed gas from condensing on the pipe wall"?*

Thanks for the careful review. Vehicle exhaust from tailpipe was first diluted by a gradient heated dilution system (6 fold) and then diluted by a unheated dilution system (5 fold). The temperature of sample flow was near indoor temperature after secondary dilution systems. The cooking fumes was collected through the kitchen ventilator, where the temperature was just a litter higher than that of indoor air. Therefore, it is enough to set the sample line temperature at 20-25°C in order to prevent freshly warmed gas from condensing on the pipe wall.

*7.L137: What are the particle backgrounds when the lamps are turned on with ozone and humidified air flowing through the Go:PAM? Simply flowing dry purified air and ozone through a dark OFR is likely insufficient to clean out the OFR between experiments that employ OH as the oxidant. In this case, the background concentrations are probably significantly underestimated because as soon as the lamps are turned on, there is the potential to generate SOA from the OH oxidation of background contaminants that are not reactive towards O3*

We appreciate the reviewer's question. We have done the $O_3$ oxidation groups during cooking experiment as Table S7 shows. There is no significant increase in OA mass when we just add O3 with UV off, comparing to those of OH oxidation groups (input O3 with UV on). We admit that $O_3$ itself indeed could influence the formation of SOA, but it is hard to study this topic in this article (we mainly consider $O_3$ as the material of OH under certain water vapor and UV level), we hope we can do more comprehensive researches in the future.

Besides, a flow reactor exposure estimator was also used in this study (Peng et al., 2016). This estimator could evaluate the potential non-OH reactions in flow reactor such as the photolysis of VOCs, the reactions with $O(^1D)$, $O(^3P)$ and $O_3$. Our results showed that non-OH reactions(including direct reaction with $O_3$)were not significant except for the photolysis of acetylacetone. But there is no acetylacetone from vehicle exhaust or cooking emission according to our measurements and previous studies. The acetylacetone was usually considered as a kind of VOCs emitted from industrial production (Ji et al., 2020). Therefore, its potential photolysis wouldn't take place during our cooking conditions, and OH reactions still played the dominant role.

**Table S7.** Comparison of primary (no O3, UV OFF), $O_3$ oxidation (certain $O_3$, UV OFF) and OH oxidation (certain $O_3$, UV ON) results during cooking experiment.

| Experiment | Input $O_3$ concentration (ppbV) | UV | OH Exposure (×10$^{10}$ molecules·cm$^{-3}$·s) | OA Concentration (μg/m3) | Standard Deviation | Relative Standard Deviation |
|---|---|---|---|---|---|---|
| Dilution Air (cooking) | - | OFF | 0 | - | - | - |
| | - | ON | 9.6 | 0.37 | 0.04 | 12% |
| Boiled Water | - | OFF | 0 | 0.04 | 0.02 | 44% |
| | - | ON | 9.6 | 0.36 | 0.12 | 32% |
| Deep-fried Chicken | - | OFF | 0 | 12.30 | 0.49 | 4% |
| | 1183 | OFF | - | 14.50 | 0.20 | 1% |
| | 1183 | ON | 9.6 | 28.29 | 2.55 | 9% |
| Shallow-fried Tofu | - | OFF | 0 | 13.56 | 0.68 | 5% |
| | 1183 | OFF | - | 14.79 | 3.25 | 22% |
| | 1183 | ON | 9.6 | 21.70 | 1.08 | 5% |
| Stir-fried Cabbage | - | OFF | 0 | 10.75 | 0.65 | 6% |
| | 1183 | OFF | - | 12.70 | 0.72 | 6% |
| | 1183 | ON | 9.6 | 18.38 | 1.65 | 9% |
| Kung Pao Chicken | - | OFF | 0 | 6.47 | 0.52 | 8% |
| | 1183 | OFF | - | / | / | / |
| | 1183 | ON | 9.6 | 11.39 | 1.25 | 11% |

Revised Text:"Except for the off-line calibration based on the decay of $SO_2$, a flow reactor exposure estimator was also used in this study (Peng et al., 2016). The OH exposures calculated by both methods showed a good correlation (Figure S1&S2). This estimator could also evaluate the potential non-OH reactions in flow reactor such as the photolysis of VOCs, the reactions with O($^1$D), O($^3$P) and $O_3$. Our results showed that non-OH reactions were not significant except for the photolysis of acetylacetone. But there is no acetylacetone from vehicle exhaust or cooking emission according to our measurements and previous studies. The acetylacetone was usually considered as a kind of VOCs emitted from industrial production (Ji et al., 2020). Therefore, its potential photolysis wouldn't take place during our cooking conditions, and OH reactions still played the dominant role. Besides, Table S7 shows the comparison of primary (no $O_3$, UV OFF), $O_3$ oxidation (certain $O_3$, UV OFF) and OH oxidation (certain $O_3$, UV ON) results during cooking experiment. There is no significant increase in OA mass when we just add $O_3$ with UV off, comparing to those of OH oxidation groups (input O3 with UV on). Overall, our Go: PAM could reasonably simulate the OH oxidation process of cooking OA in ambient."

Peng, Z., Day, D. A., Ortega, A. M., Palm, B. B., Hu, W., Stark, H., Li, R., Tsigaridis, K., Brune, W. H., and Jimenez, J. L.: Non-OH chemistry in oxidation flow reactors for the study of atmospheric chemistry systematically examined by modeling, Atmospheric Chemistry and Physics, 16, 4283-4305, 10.5194/acp-16-4283-2016, 2016.

Ji, Y., Qin, D., Zheng, J., Shi, Q., Wang, J., Lin, Q., Chen, J., Gao, Y., Li, G., and An, T.: Mechanism of the atmospheric chemical transformation of acetylacetone and its implications in night-time second organic aerosol formation, The Science of the total environment, 720, 137610, 10.1016/j.scitotenv.2020.137610, 2020

Before each experiment, all pipelines and the Go: PAM chamber were continuously flushed with purified dry air. In order to check our cleaning effect, after the cleaning procedure, we would measure

the dilution air results in the Go: PAM, and their results are similar to those dilution air blank results in Table S1, which are far below those of formal experimental groups.

**Table S1.** Comparison of results between blank and experimental groups (Dilution air and boiled water are two kinds of blank groups. The others are experimental groups.).

| Experiment | OH Exposure ($\times 10^{10}$ molecules$\cdot$cm$^{-3}\cdot$s) | OA Concentration (μg/m3) | Standard Deviation | Relative Standard Deviation |
|---|---|---|---|---|
| Dilution Air (cooking) | 0 | - | - | - |
| | 9.6 | 0.37 | 0.04 | 12% |
| Boiled Water | 0 | 0.04 | 0.02 | 44% |
| | 9.6 | 0.36 | 0.12 | 32% |
| Deep-fried Chicken | 0 | 12.30 | 0.49 | 4% |
| | 9.6 | 28.29 | 2.55 | 9% |
| Shallow-fried Tofu | 0 | 13.56 | 0.68 | 5% |
| | 9.6 | 21.70 | 1.08 | 5% |
| Stir-fried Cabbage | 0 | 10.75 | 0.65 | 6% |
| | 9.6 | 18.38 | 1.65 | 9% |
| Kung Pao Chicken | 0 | 6.47 | 0.52 | 8% |
| | 9.6 | 11.39 | 1.25 | 11% |
| Dilution Air (vehicle) | 0 | - | - | - |
| | 7.8 | 0.52 | 0.07 | 13% |
| GDI 20 km/h | 0 | 0.40 | 0.01 | 3% |
| | 7.8 | 19.68 | 1.48 | 8% |
| GDI 40 km/h | 0 | 0.41 | 0.01 | 3% |
| | 7.8 | 15.24 | 0.62 | 4% |
| GDI 60 km/h | 0 | 0.42 | 0.02 | 5% |
| | 7.8 | 23.23 | 4.00 | 17% |

*8.L159: In addition to the OFR conditions that were summarized in Tables 3-4, the actinic flux at 254 nm (or, alternatively, the ratio of O₃ measured before and after photolysis at 254 nm) is also a required input to the OFR254 OH exposure estimator. Please add this information to Tables 3-4 or describe in the text.*

As for the vehicle and cooking experiment, the photon flux at 254 nm was $4.5 \times 10^{14}$ and $2.2 \times 10^{15}$ photons$\cdot$cm$^{-2}\cdot$s$^{-1}$, respectively. These parameters have been added to SI-section S2.

Revised Text:"As for the vehicle and cooking experiment, the photon flux at 254 nm was $4.5 \times 10^{14}$ and $2.2 \times 10^{15}$ photons$\cdot$cm$^{-2}\cdot$s$^{-1}$, respectively."

*9.L161-L165: This discussion is confusing and in places incorrect. I don't understand why acetylacetone is mentioned at all if it is not present in the emissions, whereas other specific VOCs that were measured are not discussed here or anywhere else in the paper. Also, I suspect ozone is likely an important oxidant for some species, especially unsaturated fatty acids that are presumably important components of the cooking emissions as discussed in the paper. For example, for oleic acid, at OH and O₃ exposures of 2.7E11 and 5E15 molecules cm⁻³ s (OFR conditions in Line 5 of Table 4), assuming effective OH and O₃ rate coefficients of 3.5E-11 and 2.1E-15 cm³ molecule⁻¹ s⁻¹ (Renbaum et al, 2012), the estimated*

*fractional oxidative loss of oleic acid to O$_3$ is ~0.55. Thus, O$_3$ may actually be the major oxidant for several important compounds emitted from the cooking sources.*

Thanks for the constructive comments. When it comes to the reason for discussion of "acetylacetone", because our results showed that non-OH reactions were not significant except for the photolysis of acetylacetone, according to an OFR estimator (Peng et al., 2016). However, there is no acetylacetone from vehicle exhaust or cooking emission according to our measurements and previous studies. The acetylacetone was usually considered as a kind of VOCs emitted from industrial production. Therefore, its potential photolysis wouldn't take place during our cooking conditions, and OH reactions still played the dominant role.

Peng, Z., Day, D. A., Ortega, A. M., Palm, B. B., Hu, W., Stark, H., Li, R., Tsigaridis, K., Brune, W. H., and Jimenez, J. L.: Non-OH chemistry in oxidation flow reactors for the study of atmospheric chemistry systematically examined by modeling, Atmospheric Chemistry and Physics, 16, 4283-4305, 10.5194/acp-16-4283-2016, 2016.

The VOCs are measured at the inlet of Go: PAM in order to determine the OHR. We have divided them into 5 types including alkane, alkene, aromatic, O-VOCs (Oxidized VOCs, mainly included aldehyde and ketone) and X-VOCs (halogenated-VOCs) using the measurement of GC-MS (Gas Chromatography-Mass Spectrometry, GC-7890, MS-5977, Agilent Technologies Inc). The high resolution of gaseous components and their SOA or O$_3$ yields are not the main focus of this manuscript, and the detailed information of VOCs, S/I VOCs are designed to write another article, so we just list the brief result of VOCs in Table S4 here.

**Table S4.** VOCs measured by GC-MS at the inlet of Go: PAM.

| Expriment | TVOCs (ppbV) | Alkane (%) | Alkene (%) | Aromatic (%) | O-VOC (%) | X-VOC (%) |
|---|---|---|---|---|---|---|
| GDI 20 km/h | 33 | 60% | 6% | 12% | 13% | 9% |
| GDI 40 km/h | 35 | 55% | 7% | 13% | 13% | 12% |
| GDI 60 km/h | 29 | 54% | 6% | 12% | 14% | 13% |
| Deep-fried Chicken | 139 | 21% | 7% | 6% | 29% | 37% |
| Shallow-fried Tofu | 124 | 57% | 9% | 10% | 18% | 7% |
| Stir-fried Cabbage | 127 | 48% | 8% | 14% | 21% | 10% |
| Kung Pao Chicken | 189 | 64% | 8% | 11% | 5% | 13% |

Fortunately, we have done the O$_3$ oxidation groups during cooking experiment as Table S7 shows. There is no significant increase in OA mass when we just add O$_3$ with UV off, comparing to those of OH oxidation groups (input O3 with UV on). We admit that O$_3$ itself indeed could influence the formation of SOA, but it is hard to study this topic in this article (we mainly consider O$_3$ as the material of OH under certain water vapor and UV level), we hope we can do more comprehensive researches in the future.

Besides, a flow reactor exposure estimator was also used in this study (Peng et al., 2016). The OH exposures calculated by both methods showed a good correlation (Figure S1&S2). This estimator could also evaluate the potential non-OH reactions in flow reactor such as the photolysis of VOCs, the reactions with $O(^1D)$, $O(^3P)$ and $O_3$. Our results showed that non-OH reactions (including direct reaction with $O_3$)were not significant except for the photolysis of acetylacetone. But there is no acetylacetone from vehicle exhaust or cooking emission according to our measurements and previous studies. The acetylacetone was usually considered as a kind of VOCs emitted from industrial production (Ji et al., 2020). Therefore, its potential photolysis wouldn't take place during our cooking conditions, and OH reactions still played the dominant role.

**Table S7.** Comparison of primary (no O3, UV OFF), $O_3$ oxidation (certain $O_3$, UV OFF) and OH oxidation (certain $O_3$, UV ON) results during cooking experiment.

| Experiment | Input $O_3$ concentration (ppbV) | UV | OH Exposure $(\times 10^{10}$ molecules$\cdot$cm$^{-3}\cdot$s$)$ | OA Concentration ($\mu$g/m3) | Standard Deviation | Relative Standard Deviation |
|---|---|---|---|---|---|---|
| Dilution Air (cooking) | - | OFF | 0 | - | - | - |
| | - | ON | 9.6 | 0.37 | 0.04 | 12% |
| Boiled Water | - | OFF | 0 | 0.04 | 0.02 | 44% |
| | - | ON | 9.6 | 0.36 | 0.12 | 32% |
| Deep-fried Chicken | - | OFF | 0 | 12.30 | 0.49 | 4% |
| | 1183 | OFF | - | 14.50 | 0.20 | 1% |
| | 1183 | ON | 9.6 | 28.29 | 2.55 | 9% |
| Shallow-fried Tofu | - | OFF | 0 | 13.56 | 0.68 | 5% |
| | 1183 | OFF | - | 14.79 | 3.25 | 22% |
| | 1183 | ON | 9.6 | 21.70 | 1.08 | 5% |
| Stir-fried Cabbage | - | OFF | 0 | 10.75 | 0.65 | 6% |
| | 1183 | OFF | - | 12.70 | 0.72 | 6% |
| | 1183 | ON | 9.6 | 18.38 | 1.65 | 9% |
| Kung Pao Chicken | - | OFF | 0 | 6.47 | 0.52 | 8% |
| | 1183 | OFF | - | / | / | / |
| | 1183 | ON | 9.6 | 11.39 | 1.25 | 11% |

Revised Text:"Except for the off-line calibration based on the decay of $SO_2$, a flow reactor exposure estimator was also used in this study (Peng et al., 2016). The OH exposures calculated by both methods showed a good correlation (Figure S1&S2). This estimator could also evaluate the potential non-OH reactions in flow reactor such as the photolysis of VOCs, the reactions with $O(^1D)$, $O(^3P)$ and $O_3$. Our results showed that non-OH reactions were not significant except for the photolysis of acetylacetone. But there is no acetylacetone from vehicle exhaust or cooking emission according to our measurements and previous studies. The acetylacetone was usually considered as a kind of VOCs emitted from industrial production (Ji et al., 2020). Therefore, its potential photolysis wouldn't take place during our cooking conditions, and OH reactions still played the dominant role. Besides, Table S7 shows the comparison of primary (no $O_3$, UV OFF), $O_3$ oxidation (certain $O_3$, UV OFF) and OH oxidation (certain $O_3$, UV ON) results during cooking experiment. There is no significant increase in OA mass when we just add $O_3$ with UV off, comparing to those of OH oxidation groups (input O3 with UV on). Overall, our Go: PAM could reasonably simulate the OH oxidation process of cooking OA in ambient."

Peng, Z., Day, D. A., Ortega, A. M., Palm, B. B., Hu, W., Stark, H., Li, R., Tsigaridis, K., Brune, W. H., and Jimenez, J.

L.: Non-OH chemistry in oxidation flow reactors for the study of atmospheric chemistry systematically examined by modeling, Atmospheric Chemistry and Physics, 16, 4283-4305, 10.5194/acp-16-4283-2016, 2016.

Ji, Y., Qin, D., Zheng, J., Shi, Q., Wang, J., Lin, Q., Chen, J., Gao, Y., Li, G., and An, T.: Mechanism of the atmospheric chemical transformation of acetylacetone and its implications in night-time second organic aerosol formation, The Science of the total environment, 720, 137610, 10.1016/j.scitotenv.2020.137610, 2020

**10.L203: The O/C ratio is insufficient by itself to associate the mass spectra of the SOA with ambient PMF factors.**

Here, we just make a simple guess in order to introduce the following text, and a direct comparison with ambient PMF factors can be found in Section 3.4.

**11.L207 and L227-L228: These statements are too speculative, and references are made to other source characterization studies that are not directly relevant to the sources that were characterized here. Molecular speciated measurements of VOCs were performed with GC-MS as described in L168-L172 that are not discussed in the text. Those measurements should either support (or not) this interpretation of the results. And the fatty acids that are mentioned have high effective OH rate coefficients (e.g. Renbaum et al., 2012), so it is not obvious to me how the statement that "cooking produces more hardly oxidized acids" is justified.**

Thanks for the reviewer's comments. The VOCs are measured at the inlet of Go: PAM in order to determine the OHR. We have divided them into 5 types including alkane, alkene, aromatic, O-VOCs (Oxidized VOCs, mainly included aldehyde and ketone) and X-VOCs (halogenated-VOCs) using the measurement of GC-MS (Gas Chromatography-Mass Spectrometry, GC-7890, MS-5977, Agilent Technologies Inc). The high resolution of gaseous components and their SOA or $O_3$ yields are not the main focus of this manuscript, and the detailed information of VOCs, S/I VOCs are designed to write another article, so we just list the brief result of VOCs in Table S4 here.

**Table S4.** VOCs measured by GC-MS at the inlet of Go: PAM.

| Expriment | TVOCs (ppbV) | Alkane (%) | Alkene (%) | Aromatic (%) | O-VOC (%) | X-VOC (%) |
|---|---|---|---|---|---|---|
| GDI 20 km/h | 33 | 60% | 6% | 12% | 13% | 9% |
| GDI 40 km/h | 35 | 55% | 7% | 13% | 13% | 12% |
| GDI 60 km/h | 29 | 54% | 6% | 12% | 14% | 13% |
| Deep-fried Chicken | 139 | 21% | 7% | 6% | 29% | 37% |
| Shallow-fried Tofu | 124 | 57% | 9% | 10% | 18% | 7% |
| Stir-fried Cabbage | 127 | 48% | 8% | 14% | 21% | 10% |
| Kung Pao Chicken | 189 | 64% | 8% | 11% | 5% | 13% |

As the reviewer pointed out, the expression "cooking produces more hardly oxidized acids" is indeed not very appropriate. We found that the cooking SOA had lower SOA mass growth, O/C, and $f_{44}$, comparing to those of vehicle SOA. However, it is not enough to indicate that "cooking produces more hardly oxidized acids". Therefore, we have removed this inaccurate expression.

*12. L225: Is "souring" a typo? If not, I am not certain what this statement means.*

We are sorry that it was a typo. We have replaced "souring" with "increase".

*13. Figures 2-3, 7. I find it confusing/distracting to have two different photochemical age and/or two SOA:POA axis scales on the same figure.*

Thanks for the careful review. It is mainly for the beauty of the drawing, so that the data points are distributed as evenly as possible in the center of the picture

*14. Figure 6: I think that the numbered symbols represent the integer m/z values of the average vehicular and cooking exhaust AMS spectra, but this should be made clearer in the legend or the caption.*

As the reviewer suggested, we have explained them in the caption.

Revised text "Figure 6. Average mass spectra of OA from two urban lifestyle sources. The numbered symbols represent the m/z values with relatively large fractions. The gray symbols represent the fragments mainly come from hydrocarbon-like fragments and the green symbols represent those mainly come from oxygen-containing fragments."

*References provided by the reviewer*

*L. Renbaum-Wolff and G. D. Smith, "Virtual Injector" Flow Tube Method for Measuring Relative Rates Kinetics of Gas-Phase and Aerosol Species, J. Phys. Chem. A, 116, 6664−6674, dx.doi.org/10.1021/jp303221w, 2012.*

*Pagonis, D., Krechmer, J. E., de Gouw, J., Jimenez, J. L., and Ziemann, P. J.: Effects of gas–wall partitioning in Teflon tubing and instrumentation on time-resolved measurements of gas-phase organic compounds, Atmos. Meas. Tech., 10, 4687–4696, https://doi.org/10.5194/amt-10-4687-2017, 2017.*

*References in Reply*

Ji, Y., Qin, D., Zheng, J., Shi, Q., Wang, J., Lin, Q., Chen, J., Gao, Y., Li, G., and An, T.: Mechanism of the atmospheric chemical transformation of acetylacetone and its implications in night-time second organic aerosol formation, The Science of the total environment, 720, 137610, 10.1016/j.scitotenv.2020.137610, 2020.

Peng, Z., Day, D. A., Ortega, A. M., Palm, B. B., Hu, W., Stark, H., Li, R., Tsigaridis, K., Brune, W. H., and Jimenez, J. L.: Non-OH chemistry in oxidation flow reactors for the study of atmospheric chemistry systematically examined by modeling, Atmospheric Chemistry and Physics, 16, 4283-4305, 10.5194/acp-16-4283-2016, 2016.

Xu, W., Han, T., Du, W., Wang, Q., Chen, C., Zhao, J., Zhang, Y., Li, J., Fu, P., Wang, Z., Worsnop, D. R., and Sun, Y.: Effects of Aqueous-Phase and Photochemical Processing on Secondary Organic Aerosol Formation and Evolution in Beijing, China, Environmental science & technology, 51, 762-770, 10.1021/acs.est.6b04498, 2017.

---

## Referee Report (RR1)

1. The authors' reply to my comment #1 added information about the tubing materials and the Go:PAM reactor geometry, but information about the inlet tubing length, diameter, and residence time were not given, and are still needed.

2. The reply to my comment #2 concedes my point that the different residence times complicate direct comparisons between cooking and vehicle exhaust measurements. This needs to be stated in the methods section, so that readers understand the importance of maintaining a fixed residence time and apply it in their own work. Similarly, the concluding section would benefit from a sentence or two that is more committal to future better-quality OFR experiments than "we really hope we could strictly control the temperature, RH, and other conditions". The authors argue that that the relative humidity, although different between in the two sources, is low enough in both cases to minimally influence heterogenous chemistry. This is a fair point, and one that should be added to the revised manuscript.

3. In response to my comment #3, the authors added fractional contributions of alkanes, alkenes, aromatics, O-VOCs, and X-VOCs to the measured VOC emissions from each source, along with kOH values for the major components of each of those compound classes. This added information is appreciated and is likely sufficient to give the reader a general idea of which compounds contribute to the calculated OH reactivity values. However, in my opinion, the discussion of which emitted compounds contribute to the measured SOA formation is an important outcome of this study whose discussion remains unclear/incomplete. For example, the SOA/POA ratio was ~100 for GDI emissions and ~2 for cooking emissions, but only 29-35 ppb VOCs were speciated in GDI emissions and 124-189 ppb VOCs were speciated in cooking emissions. It is clear from these numbers that the aggregate SOA yields of cooking VOCs are lower than GDI VOCs. It is not clear how much of the observed SOA can be explained from the speciated VOCs (and laboratory SOA yields obtained in chamber studies), and whether an unresolved complex mixture of (I)VOCs is plausibly an important class of SOA precursors, particularly for the GDI emissions. To increase the scientific significance of this manuscript, I feel strongly that additional analysis needs to be done in this regard, such that we can gain insight into contributions of measured VOCs to observed SOA formation in both sources. See, for example, Figure 3 of Liao et al., ES&T, 2021, reproduced below for reference – it should be possible to perform a similar analysis from the data obtained here.

[Figure]

**Figure 3.** Mean observed $OA_{AE}$ corresponding to the equivalent photochemical age of $1.5 \pm 0.2$ days and the predicted $OA_{AE}$ contributed from OH oxidation of precursors measured with PTR-QiTOF. Error bars show the standard deviations of the data for each case with the sample size shown as $n$ below. VOC categories are described in the Materials and Methods section. Selected OVOCs include $CH_3OH$, $C_2H_6O$, $C_4H_8O$, $C_4H_8O_2$, $C_5H_6O$, $C_6H_6O$, and $C_8H_{10}O$.

4.  OK

5.  The addition of Table S7 is useful and appreciated. However, the discussion about non-OH artifacts misses the point : ozonolysis is (likely) important for unsaturated fatty acids that were probably emitted from the cooking sources, and perhaps measured (e.g. oleic acid parent and dehydration peaks at AMS m/z = 282 and 264), but have not been analyzed. At the least, a couple sentences should be added to the discussion indicating that contributions from ozonolysis of unsaturated fatty acids to SOA/OPOA are not constrained in the data that has been analyzed but may be important.

6.  I suggest paraphrasing this response and adding it to the revised manuscript.

7.  This response does not address my comment. Cleaning the (dark) Go:PAM by flowing dry air and ozone through it (very likely) does not remove all of the residual organics that are reactive towards OH but not O3. Thus, the next time the lights are turned on to make OH, some of the SOA formed is almost certainly associated with residual organics that were not sufficiently removed from the previous experiment by only flowing dry carrier gas + ozone through the dark OFR between experiments. As far as I can tell, and based on my own experience, it is impossible to rule this out the way the experiments were performed. This is a flaw in the cleaning protocol that has to be expressed in the methods section - i.e. that the SOA that is formed in any experiment with OH as the oxidant represents an upper limit because the background is not well constrained. This should be directly addressed in future studies by adding humidity to the carrier gas and turning on the lights during the OFR cleanout stage.

8.  Thank you.

9.  I disagree that discussion of acetylacetone is useful when there is no acetylacetone in the emissions. Please focus the discussion of non-OH artifacts on the alkanes, alkenes, aromatics, O-VOCs, and X-VOCs that were measured, and the co-emitted compounds that were emitted and (likely) measured but not analyzed (e.g. fatty acids).

10. No additional comments

11. See response to comment #3. The authors argue that it is beyond the scope of this paper to discuss the SOA formation potential of individual measured VOCs. I disagree – in my opinion, that is one

of the potentially most novel results for which data is already available. Further analysis and discussion is required (e.g. Liao et al., 2021).

12. Thank you.

13. Figure 2 would be much easier (for me) to read if it were split into two panels or a single "split" y-axis, one with an SOA/POA axis ranging from 0.4 to 4, the other with an SOA/POA axis ranging from 40 to 200. The maximum photochemical ages (~3 and 6 days) are close enough that, in my opinion, they can both be plotted satisfactorily on the same x-axis, here and also in Figure 3.

14. Thank you.

**References**

Liao, K., Q. Chen, Y. Liu, Y. J. Li, A. T. Lambe, T. Zhu, R.-J. Huang, Y. Zheng, X. Cheng, R. Miao, G. Huang, R. B. Khuzestani and T. Jia. Secondary Organic Aerosol Formation of Fleet Vehicle Emissions in China: Potential Seasonality of Spatial Distributions. Environmental Science & Technology, https://doi.org/10.1021/acs.est.0c08591, 2021.

---

## Author Response (AR2)

**Comments From Reviewer 2**

We appreciate the constructive comments from the reviewer on this manuscript. We have answered the reviewer's questions point to point in the following paragraphs (the texts italicized are the comments, the texts indented are the responses, and the texts in blue are revised parts in the new manuscript). In addition, all changes made are marked in the revised manuscript.

*1.The authors' reply to my comment #1 added information about the tubing materials and the Go:PAM reactor geometry, but information about the inlet tubing length, diameter, and residence time were not given, and are still needed.*

Thanks for the reviewer's response. In order to make the article easier to follow, we have supplied the information about the inlet tubing length, diameter, and residence time in SI.

"**Revised text in Section S1:** From the sampling port at the source (cooking and vehicle) to the inlet of HR-ToF-AMS, the 3/8 inch (inner diameter was 6 mm) stainless steel tubes were totally 7 meters long and the corresponding residence time was 4.9 s. There were 5 meters long from the sampling port to the Go: PAM, and the flow rate was 5.5 L/min (HR-Tof-AMS and other instruments jointly determined the flow rate). There were 2 meters long from the Go: PAM to the HR-ToF-AMS, and the flow rate was 1 L/min (HR-ToF-AMS and its drainage system determined the flow rate)."

*2.The reply to my comment #2 concedes my point that the different residence times complicate direct comparisons between cooking and vehicle exhaust measurements. This needs to be stated in the methods section, so that readers understand the importance of maintaining a fixed residence time and apply it in their own work. Similarly, the concluding section would benefit from a sentence or two that is more committal to future better-quality OFR experiments than "we really hope we could strictly control the temperature, RH, and other conditions". The authors argue that that the relative humidity, although different between in the two sources, is low enough in both cases to minimally influence heterogenous chemistry. This is a fair point, and one that should be added to the revised manuscript.*

Thanks for the reviewer's suggestion. We have supplied necessary expressions in Method and Conclusion sections.

"**Revised text in Method 2.3.2:** "The Go: PAM conditions for vehicle and cooking experiments could be seen in Table 3 and Table 4, respectively. Their experiment conditions (such as residence time and RH) were not completely the same because of the inherent difference and experimental design between two sources. Whereas, some comparisons could be still analyzed in the similar OH exposure, and their RH conditions were both low where photochemical oxidations instead

of aqueous-phase processing dominated the chemical evolution process (Xu et al., 2017)."

"**Revised text in Conclusion:** "There are some uncertainties of our Go: PAM simulation. We focused more on the photochemical oxidation of SOA under low RH levels, but aqueous-phase processing at high RH levels may also have impacts to SOA production. In the future, it'll be better to strictly control the RH, high/low $NO_x$ or $SO_2$, additional inorganic seeds, and so forth, to deeply investigate how the aerosol ages as a function of equivalent days of atmospheric oxidation."

*Reference:*

*Xu, W., Han, T., Du, W., Wang, Q., Chen, C., Zhao, J., Zhang, Y., Li, J., Fu, P., Wang, Z., Worsnop, D. R., and Sun, Y.: Effects of Aqueous-Phase and Photochemical Processing on Secondary Organic Aerosol Formation and Evolution in Beijing, China, Environmental science & technology, 51, 762-770, 10.1021/acs.est.6b04498, 2017.*

*3. In response to my comment #3, the authors added fractional contributions of alkanes, alkenes, aromatics, O-VOCs, and X-VOCs to the measured VOC emissions from each source, along with kOH values for the major components of each of those compound classes. This added information is appreciated and is likely sufficient to give the reader a general idea of which compounds contribute to the calculated OH reactivity values. However, in my opinion, the discussion of which emitted compounds contribute to the measured SOA formation is an important outcome of this study whose discussion remains unclear/incomplete. For example, the SOA/POA ratio was ~100 for GDI emissions and ~2 for cooking emissions, but only 29-35 ppb VOCs were speciated in GDI emissions and 124-189 ppb VOCs were speciated in cooking emissions. It is clear from these numbers that the aggregate SOA yields of cooking VOCs are lower than GDI VOCs. It is not clear how much of the observed SOA can be explained from the speciated VOCs (and laboratory SOA yields obtained in chamber studies), and whether an unresolved complex mixture of (I)VOCs is plausibly an important class of SOA precursors, particularly for the GDI emissions. To increase the scientific significance of this manuscript, I feel strongly that additional analysis needs to be done in this regard, such that we can gain insight into contributions of measured VOCs to observed SOA formation in both sources. See, for example, Figure 3 of Liao et al., ES&T, 2021, reproduced below for reference – it should be possible to perform a similar analysis from the data obtained here.*

Thanks for the constructive comments. The reviewer pointed out that "*For example, the SOA/POA ratio was ~100 for GDI emissions and ~2 for cooking emissions, but only 29-35 ppb VOCs were speciated in GDI emissions and 124-189 ppb VOCs were speciated in cooking emissions.*" In fact, The GDI emission underwent a larger dilution ratio than cooking emission did, so that the VOCs from GDI is not far lower than those from cooking.

We agree with the reviewer's comment that the gaseous results are important. In this article, we just used

simple VOCs results from GC-MS in order to estimate the OHR. If we need to do a further analysis, more sophisticated results are necessary. To be frank, we have measured the VOCs and S/I VOCs by 2D-GC-MS, VOCUS and CIMS during Go: PAM experiments, but their sampling, quantification and data analysis are so complicated and different from particle phase, that our group hope to write individual articles to show our findings in detail. As following figures show, here are some initial results of cooking experiment (unpublished):

**Gas-phase**

- Significant difference between four dishes in gas-phase total S/I VOCs (p < 0.05)
- Kung Pao chicken & Pan-fried tofu: large proportion of aromatics
- Furans were abundant in gas-phase oxygenated compounds (> 6%)

[Figure]

**Particle-phase**

- NO significant difference between four dishes in particle-phase total S/IVOCs (p > 0.8)
- Oxygenated compounds were the most abundant (>40%), including acids, furans, amides and esters

[Figure]

**SOA formation**

- 26% - 68% SOA could be explained from oxidation of VOCs, IVOCs, SVOCs precursors

[Figure]

This resubmitted manuscript focused more on particle phase (based on HR-ToF-AMS), especially its SOA growth factor, O/C, oxidation pathway and mass spectra. We have compared cooking and vehicle SOA mass spectra with ambient SOA spectra in Beijing (Discussion 3.4). In the future, the cooking and vehicle SOA mass spectra could be used as the input mass spectra in ME-2, in order to estimate the mass concentration of cooking SOA and vehicle SOA, and then study their chemical evolutions in ambient. We are designing another article to study the application of our SOA mass spectra (cooking and vehicle) in urban areas. As following figures show, here is one practice result in Beijing (unpublished).

[Figure]

Timeseries of OA factors and good correlations with relevant species

[Figure]

Reasonable diurnal variations of OA factors

Therefore, the results in this article are valuable and could be used to do further estimation in ambient. We regarded this article as a beginning of our "source secondary simulation" research. We will try our best to give more informative results in other new articles as soon as possible. Thanks for the reviewer's helpful advice again.

*4.OK*

Thanks for the helpful advice.

*5.The addition of Table S7 is useful and appreciated. However, the discussion about non-OH artifacts misses the point : ozonolysis is (likely) important for unsaturated fatty acids that were probably emitted from the*

*cooking sources, and perhaps measured (e.g. oleic acid parent and dehydration peaks at AMS m/z = 282 and 264), but have not been analyzed. At the least, a couple sentences should be added to the discussion indicating that contributions from ozonolysis of unsaturated fatty acids to SOA/OPOA are not constrained in the data that has been analyzed but may be important.*

We agree with the reviewer that ozonolysis is (likely) important for unsaturated fatty acids that were probably emitted from the cooking sources, but we didn't find the significant effect of ozonolysis in our experiment. This may be because the residence time is too short for ozonolysis reactions. Besides, Vesna *et al.* found that the heterogeneous reaction of ozone with oleic acid aerosol particles was influenced by humidity and reaction time in an aerosol flow reactor. We have added necessary expressions in Method and Conclusion section.

"**Revised text in Method 2.3.2: "**It is found that the heterogeneous reaction of ozone with oleic acid aerosol particles was influenced by humidity and reaction time in an aerosol flow reactor (Vesna et al., 2009). Therefore, non-OH reactions, such as the ozonolysis of unsaturated fatty acids, may also be important in forming SOA, which missed specific designs in our experiment."

"**Revised text in Conclusion:** "Moreover, contribution of ozonolysis to SOA formation should be individually studied in further research."

*6.I suggest paraphrasing this response and adding it to the revised manuscript.*

We have added this response in Method section as the review suggested.

"**Revised text in Method 2.1:** Vehicle exhaust from tailpipe was first diluted by a gradient heated dilution system (6 fold) and then diluted by an unheated dilution system (5 fold). The temperature of sample flow was near indoor temperature (20-25°C) after secondary dilution systems. The cooking fumes was collected through the kitchen ventilator, where the temperature was similar to that of indoor air."

*7.This response does not address my comment. Cleaning the (dark) Go:PAM by flowing dry air and ozone through it (very likely) does not remove all of the residual organics that are reactive towards OH but not O3. Thus, the next time the lights are turned on to make OH, some of the SOA formed is almost certainly associated with residual organics that were not sufficiently removed from the previous experiment by only flowing dry carrier gas + ozone through the dark OFR between experiments. As far as I can tell, and based on my own experience, it is impossible to rule this out the way the experiments were performed. This is a flaw in the cleaning protocol that has to be expressed in the methods section - i.e. that the SOA that is formed in any experiment with OH as the oxidant represents an upper limit because the background is not well constrained. This should be directly addressed in future studies by adding humidity to the carrier gas and turning on the*

*lights during the OFR cleanout stage.*

Thanks for the reviewer's careful reviews on technical details of our experiment. As reviewer suggested, it is recommended to add humidity to the carrier gas and turn on the lights during the OFR cleanout stage, in order to minimize the background concentration in Go: PAM. We feel grateful and will remember this important point in the future.

As for our experiments, before each experiment, all pipelines and the Go: PAM chamber were continuously flushed with purified dry air, until the concentrations were minimal (just like blank groups in Table S1) when the UV was on or off. We agree with that the SOA formed represented an upper limit because the background was not totally clean, but its influence is minimal according to the comparison of blank and experimental groups in Table S1. We have supplied necessary expressions in Method and Conclusion section.

**Table S1.** Comparison of results between blank and experimental groups (Dilution air and boiled water are two kinds of blank groups. The others are experimental groups).

| Experiment | OH Exposure ($\times 10^{10}$ molecules·cm$^{-3}$·s) | OA Concentration (μg/m3) | Standard Deviation | Relative Standard Deviation |
|---|---|---|---|---|
| Dilution Air (cooking) | 0 | - | - | - |
| | 9.6 | 0.37 | 0.04 | 12% |
| Boiled Water | 0 | 0.04 | 0.02 | 44% |
| | 9.6 | 0.36 | 0.12 | 32% |
| Deep-fried Chicken | 0 | 12.30 | 0.49 | 4% |
| | 9.6 | 28.29 | 2.55 | 9% |
| Shallow-fried Tofu | 0 | 13.56 | 0.68 | 5% |
| | 9.6 | 21.70 | 1.08 | 5% |
| Stir-fried Cabbage | 0 | 10.75 | 0.65 | 6% |
| | 9.6 | 18.38 | 1.65 | 9% |
| Kung Pao Chicken | 0 | 6.47 | 0.52 | 8% |
| | 9.6 | 11.39 | 1.25 | 11% |
| Dilution Air (vehicle) | 0 | - | - | - |
| | 7.8 | 0.52 | 0.07 | 13% |
| GDI 20 km/h | 0 | 0.40 | 0.01 | 3% |
| | 7.8 | 19.68 | 1.48 | 8% |
| GDI 40 km/h | 0 | 0.41 | 0.01 | 3% |
| | 7.8 | 15.24 | 0.62 | 4% |
| GDI 60 km/h | 0 | 0.42 | 0.02 | 5% |
| | 7.8 | 23.23 | 4.00 | 17% |

"**Revised text in Method 2.2:** "Before each experiment, all pipelines and the Go: PAM chamber were continuously flushed with purified dry air, until the concentrations were minimal (just like blank groups in Table S1) when the UV was on or off. The SOA formed in each experiment represented the upper limit due to the presence of background concentration."

"**Revised text in Conclusion:** "Besides, it is recommended to add humidity to the carrier gas and turn on the lights during the OFR cleanout stage, in order to minimize the background concentration in the Go: PAM."

*8. Thank you.*

Thanks for the careful review.

*9.I disagree that discussion of acetylacetone is useful when there is no acetylacetone in the emissions. Please focus the discussion of non-OH artifacts on the alkanes, alkenes, aromatics, OVOCs, and X-VOCs that were measured, and the co-emitted compounds that were emitted and (likely) measured but not analyzed (e.g. fatty acids).*

Thanks for the reviewer's comments. We have removed the discussion of acetylacetone.

*10.No additional comments*

Thanks for the constructive comments.

*11.See response to comment #3. The authors argue that it is beyond the scope of this paper to discuss the SOA formation potential of individual measured VOCs. I disagree – in my opinion, that is one of the potentially most novel results for which data is already available. Further analysis and discussion is required (e.g. Liao et al., 2021).*

Thanks for the constructive comments. This resubmitted manuscript focused more on particle phase, especially its SOA growth factor, O/C, oxidation pathway and mass spectra. We have compared cooking and vehicle SOA mass spectra with ambient SOA spectra in Beijing (Discussion 3.4). In the future, the cooking and vehicle SOA mass spectra could be used as the input mass spectra in ME-2, in order to estimate the mass concentration of cooking SOA and vehicle SOA, and then study their chemical evolutions in ambient. We are designing another article to study the application of our SOA mass spectra (cooking and vehicle) in urban areas. As following figures show, here is one practice in Beijing (unpublished):

[Figure]

Timeseries of OA factors and good correlations with relevant species

[Figure]

Reasonable diurnal variations of OA factors

Therefore, the results in this article are valuable and could be used to do further estimation in ambient. We regarded this article as a beginning part of our "source secondary simulation" research. We will try our best to give more informative results in other new articles as soon as possible.

We agree with the reviewer's comment that the gaseous results are important. In this article, we just use simple VOCs results from GC-MS in order to estimate the OHR. As for the further measurement and analysis results of gaseous phase, we have regarded it as another independent and attractive part. In fact, we have measured the VOCs and S/I VOCs by 2D-GC-MS, VOCUS and CIMS during Go: PAM experiments, but their sampling, quantification and data analysis are so complicated and different from particle phase, that we need to write individual articles to show our findings. Thanks for the reviewer's helpful advice again. As the following figures show, here are some initial results of cooking experiment (unpublished). We really hope our findings would benefit other researchers' studies in the future.

**Gas-phase**

- Significant difference between four dishes in gas-phase total S/I VOCs ($p < 0.05$)
- Kung Pao chicken & Pan-fried tofu: large proportion of aromatics
- Furans were abundant in gas-phase oxygenated compounds ($> 6\%$)

[Figure]

**Particle-phase**

- NO significant difference between four dishes in particle-phase total S/IVOCs (p > 0.8)
- Oxygenated compounds were the most abundant (>40%), including acids, furans, amides and esters

[Figure]

**SOA formation**

- 26% - 68% SOA could be explained from oxidation of VOCs, IVOCs, SVOCs precursors

12. *Thank you.*

   Thanks for the correction.

13. *Figure 2 would be much easier (for me) to read if it were split into two panels or a single "split" yaxis, one with an SOA/POA axis ranging from 0.4 to 4, the other with an SOA/POA axis ranging from 40 to 200. The maximum photochemical ages (~3 and 6 days) are close enough that, in my opinion, they can both be plotted satisfactorily on the same x-axis, here and also in Figure 3.*

   As the reviewer advised, we have modified the Figure 2 and Figure 3 into the same x-axis.

[Figure]

**Figure 2.** Secondary mass growth potentials for two urban lifestyle SOA. The SMPS-1 determined the mass concentration of POA, while the SMPS-2 determined the mass concentration of aged OA, and their mass difference could be regarded as the SOA. The average data and standard deviation bars are shown in the figure.

[Figure]

**Figure 3.** Evolution of O:C molar ratio for two urban lifestyle OA. The O:C molar ratios are determined by HR-Tof-AMS. The average data and standard deviation bars at each gradient are shown in the figure.

*14.Thank you.*

Thanks for the positive reply.

*References*

*Liao, K., Q. Chen, Y. Liu, Y. J. Li, A. T. Lambe, T. Zhu, R.-J. Huang, Y. Zheng, X. Cheng, R. Miao, G. Huang, R. B. Khuzestani and T. Jia. Secondary Organic Aerosol Formation of Fleet Vehicle Emissions in China: Potential Seasonality of Spatial Distributions. Environmental Science & Technology, https://doi.org/10.1021/acs.est.0c08591, 2021.*

We have added this article in Introduction section. Thanks for the positive reply.

"**Revised text in Introduction:** "$NO_x$ levels may greatly influence the chemical evolution of vehicle SOA, and its $NO_x$/VOCs values are often strongly dependent on the sampling time and place in urban areas (Zhan et al., 2021;Wei et al., 2014). It is found that the photochemical ages for maximum SOA production under high-$NO_x$ levels were lower than those under low-$NO_x$ levels among OFR simulations (Liao et al., 2021)."

**Comments From Reviewer 3**

We appreciate the constructive comments from the reviewer on this manuscript. We have answered the reviewer's questions point to point in the following paragraphs (the texts italicized are the comments, the texts indented are the responses, and the texts in blue are revised parts in the new manuscript). In addition, all changes made are marked in the revised manuscript.

*1.Responses to Comment 2-1 and 2-6: first I do not understand the response to Comment 2-6. Did the authors mean that because the fume was only a little warmer than indoor air, it had already cooled down and condensable gases condensed onto particles? If so, it is unclear to me what role pipe heating really played and the phrase "prevent freshly warmed gas from condensing on the pipe wall" does not seem to be appropriate; otherwise, I am not convinced that pipe heating was sufficient to prevent wall losses/tubing delay of S/IVOCs in the pipe. Obviously, pipe heating could help substantially. It is however unclear whether this measure can reduce wall/tubing effects in the pipe to minor/negligible level unless the authors perform a more quantitative analysis e.g. based on Pagonis et al. (2017).*

> Thanks for the reviewer's patience on our manuscript. To be exact, the fume was hot after emission, but was only a bit warmer than the indoor air after our dilution system. As for vehicle experiment, vehicle exhaust from tailpipe was first diluted by a gradient heated dilution system (6 fold) and then diluted by a unheated dilution system (5 fold). The temperature of sample flow was near indoor temperature (20-25℃) after secondary dilution systems. As for cooking experiment, the cooking fumes was collected through the kitchen ventilator, where the temperature was just similar to that of indoor temperature (20-25℃).

> Compared to the condensation caused by instantaneous temperature difference, our gradient heated dilution system and slight temperature difference (between emission and measurement) could decrease the loss of gaseous compounds. In this article, the main instrument is HR-Tof-AMS which focused on the particle phase, so our sampling tubes were made of stainless steel rather than Teflon tubes (Pagonis et al discussed the wall loss of Teflon tubes). The reviewer 2 has accepted our response and suggested us to supply necessary description in the Method section as following. Thanks for the reviewers again.

> "**Revised text in Method 2.1:** "Vehicle exhaust from tailpipe was first diluted by a gradient heated dilution system (6 fold) and then diluted by an unheated dilution system (5 fold). The temperature of sample flow was near indoor temperature (20-25℃) after secondary dilution systems. The cooking fumes was collected through the kitchen ventilator, where the temperature was similar to that of indoor air."

S/I VOCs were indeed partly lost in pipelines, and its sampling and quantification are really hard and challenging, which was actually one of uncertainties of our study. We have added this uncertainty in the Conclusion section.

"**Revised text in Conclusion:** "S/I VOCs may play important roles in forming SOA but were indeed partly lost in pipelines, and its sampling and quantification are really hard and challenging, which needs more sophisticated experimental design."

*2.Response to Comment 2-9: the observation that oleic acid ozonolysis made little SOA in Go:PAM does not imply that this reaction cannot produce SOA significantly in the atmosphere. Ozonolysis of oleic acid breaks its long chain and produces higher-volatility products than its oxidation by OH does. It does not surprise me that the ozonolysis products did not condense onto particles during the very short residence time of Go:PAM. However, in the atmosphere, the O3:OH ratio is even higher than in Go:PAM and the products of e.g. oleic acid ozonolysis, particularly organic peroxy radicals, have enough time to undergo several generations of autoxidation to become of sufficiently low volatility and condense. Therefore, the (lack of) contribution of ozonolysis to SOA formation in Go:PAM should be seriously discussed.*

We agree with the reviewer that ozonolysis is (likely) important for unsaturated fatty acids that were probably emitted from the cooking sources, but we didn't find the significant effect of ozonolysis in our experiment. This may be because the residence time is too short for ozonolysis reactions. Besides, Vesna et al. found that the heterogeneous reaction of ozone with oleic acid aerosol particles was influenced by humidity and reaction time in an aerosol flow reactor. We have added necessary expressions in Method and Conclusion section.

"**Revised text in Method 2.3.2:** "It is found that the heterogeneous reaction of ozone with oleic acid aerosol particles was influenced by humidity and reaction time in an aerosol flow reactor (Vesna et al., 2009). Therefore, non-OH reactions, such as the ozonolysis of unsaturated fatty acids, may also be important in forming SOA, which missed specific designs in our experiment."

"**Revised text in Conclusion:** "Moreover, contribution of ozonolysis to SOA formation should be individually studied in further research."

*References:*

*Vesna, O., Sax, M., Kalberer, M., Gaschen, A., and Ammann, M.: Product study of oleic acid ozonolysis as function of humidity, Atmospheric Environment, 43, 3662-3669, 10.1016/j.atmosenv.2009.04.047, 2009.*

*3.In addition, I have 2 major concerns about this paper:- Liao et al. (2021) did OFR experiments of SOA formation from on-road vehicle emissions in Beijing. They kept their experimental conditions high-NOx and*

*an earlier and relatively low peak of SOA formation (as a function of photochemical age), compared to literature studies of this kind, which were all conducted under low-NOx conditions in OFRs. The peak SOA age in Liao et al. (2021) was only ~1 d, significantly lower than in this study (~2 d). High-NOx pathways in organic peroxy radical chemistry allow fragmentation of organics to occur at lower degree of oxidation than low-NOx pathways. As a result, the compositions of SOA formed through these 2 types of pathways may be significantly different. I thus have doubts whether the SOA formed in Go:PAM in this study is sufficiently representative of SOA formed in urban areas, where NOx is usually high. If the SOA was not representative, the analysis in the triangle plot and the VK diagram, PMF etc. in the paper did not bring much insight. I believe that the authors should justify that the SOA in their experiments were representative of SOA in urban atmospheres.*

We appreciate the reviewer's reasonable concern and careful review. We agree with that different $NO_x$/VOCs conditions may have significant influence on SOA formation. In urban areas, high/low $NO_x$ conditions may appear alternately, it strongly depends on the place and time of observation. As the review mentioned, Liao et al (2021) conducted the OFR simulation on a busy road of Beijing under high-$NO_x$ condition. However, Wei et al (2014) found that low-$NO_x$ (High VOCs) phenomenon around a refinery in Beijing (as shown in the following figure). A busy road may emit more $NO_x$, and a refinery may emit more VOCs. High/low $NO_x$ (VOCs/$NO_x$) greatly depends on the measurement place.

[Figure]

**Fig. 5**  Hourly mean NOx concentrations in daytime around the refinery.

[Figure]

**Fig. 6**  TVOCs concentrations at the three sites at different hours. The point at one line meant the simultaneous observations at three sites on a monitor city.

Moreover, Zhan et al (2021)'s field study in Beijing showed the timeseries of VOCs and $NO_x$. The concentration of VOCs was underestimated, because only 57 NMHCs standard materials were used to

quantify the C2-C12 NMHC measured by GC-FID. As the figure below shows, the concentration of $NO_x$ was not obviously higher than VOCs all the time, and the concentration of VOCs was not obviously higher than $NO_x$ all the time as well. The $NO_x$/VOCs condition is dynamic, it depends greatly on the measurement time.

[Figure]

Overall, high/low $NO_x$ condition or other complicated atmospheric conditions are all likely to occur in urban areas, and it greatly depends on the place and time of observation.

There are indeed uncertainties in our laboratory simulation, when we only conducted vehicle experiment under moderate-$NO_x$ condition (the concentration of $NO_x$ is similar to that of VOCs) and cooking experiment under low-$NO_x$ condition (the concentration of $NO_x$ is far lower than that of VOCs). Hence, further researches are still needed. We have added the expression of this uncertainties in Conclusion and Introduction section.

"**Revised text in Conclusion:** "In the future, it'll be better to strictly control the temperature, RH, high/low $NO_x$ or $SO_2$, additional inorganic seeds, and so forth, to deeply investigate how the aerosol ages as a function of equivalent days of atmospheric oxidation.""

"**Revised text in Introduction:** "$NO_x$ levels may greatly influence the chemical evolution of vehicle SOA, and its $NO_x$/VOCs values are often strongly dependent on the sampling time and place in urban areas (Zhan et al., 2021; Wei et al., 2014). It is found that the photochemical ages for maximum SOA production under high-$NO_x$ levels were lower than those under low-$NO_x$ levels among OFR simulations (Liao et al., 2021).""

*References:*

*Wei, W., Cheng, S., Li, G., Wang, G., and Wang, H.: Characteristics of ozone and ozone precursors (VOCs and NOx) around a petroleum refinery in Beijing, China, Journal of Environmental Sciences, 26, 332-342, 10.1016/s1001-0742(13)60412-x, 2014.*

*Zhan, J., Feng, Z., Liu, P., He, X., He, Z., Chen, T., Wang, Y., He, H., Mu, Y., and Liu, Y.: Ozone and SOA formation potential based on photochemical loss of VOCs during the Beijing summer, Environmental pollution, 285, 117444, 10.1016/j.envpol.2021.117444, 2021.*

*Liao, K., Chen, Q., Liu, Y., Li, Y. J., Lambe, A. T., Zhu, T., Huang, R.-J., Zheng, Y., Cheng, X., Miao, R., Huang, G., Khuzestani, R. B., and Jia, T.: Secondary Organic Aerosol Formation of Fleet Vehicle Emissions in China: Potential Seasonality of Spatial Distributions, Environ. Sci. Technol., 55, 7276–7286, 2021.*

*4.In a number of places in the paper, the discussions about SOA formation from cooking emissions seemed to be based on heterogeneous oxidation of POA. While I understand that this pathway may play some role, it is slow enough to be minor at photochemical ages of 1-3 days compared to gas-phase oxidation of cooking emissions, a large fraction of which is S/IVOC with C=C bonds. The authors should discuss how cooking SOA is formed from S/IVOC oxidation in the gas phase.*

Thanks for the constructive comments. This resubmitted manuscript focused more on particle phase, especially its SOA growth factor, O/C, oxidation pathway and mass spectra. We have compared cooking and vehicle SOA mass spectra with ambient SOA spectra in Beijing (Discussion 3.4). In the future, the cooking and vehicle SOA mass spectra could be used as the input mass spectra in ME-2, in order to estimate the mass concentration of cooking SOA and vehicle SOA, and then study their chemical evolution in ambient. We are designing another article to study the application of our SOA mass spectra (cooking and vehicle) in urban areas. As following figures show, here is one practice in Beijing(unpublished):

[Figure]

Timeseries of OA factors and good correlations with relevant species

[Figure]

Reasonable diurnal variations of OA factors

Therefore, the results in this article are valuable and could be used to do further estimation in ambient. We regarded this article as a beginning of our "source secondary simulation" research. We will try our best to give more informative results in other new articles as soon as possible.

We agree with the reviewer's comment that the gaseous results are important. In this article, we just use simple VOCs results from GC-MS in order to estimate the OHR. As for the further measurement and analysis results of gaseous phase, we have regarded it as another independent and attractive part. In fact, we have measured the VOCs and S/I VOCs by 2D-GC-MS, VOCUS and CIMS during Go: PAM experiments, but their sampling, quantification and data analysis are so complicated and different from particle phase, that we need to write individual articles to show our findings. Thanks for the reviewer's helpful advice again. As the following figures show, here are some initial results of cooking experiment (unpublished). We really hope our findings would benefit other researcher's studies in the future.

**Gas-phase**

- Significant difference between four dishes in gas-phase total S/I VOCs (p < 0.05)
- Kung Pao chicken & Pan-fried tofu: large proportion of aromatics
- Furans were abundant in gas-phase oxygenated compounds (> 6%)

[Figure]

**Particle-phase**

- NO significant difference between four dishes in particle-phase total S/IVOCs (p > 0.8)
- Oxygenated compounds were the most abundant (>40%), including acids, furans, amides and esters

5.*Response to Comment 1-9: I agree that Manchester is not a megacity. But this would be better determined by the total population of its metro area (~3 M) than by that of the city proper (~0.5 M).*

We agree with this determination. Thanks for the suggestion.

"**Revised text in Introduction:** "take the megacity (total population of its metro area is more than 3 M) for example"

6.*Response to Comment 2-2: to me, the authors seemed to misunderstand the Reviewer's point here. The Reviewer likely meant that low humidity would lead to highly viscous particles, which usually undergo slower heterogeneous oxidation than less viscous particles.*

We agree with that low humidity would lead to highly viscous particles, which usually undergo slower heterogeneous oxidation than less viscous particles. This is the uncertainty of our experiment. Therefore, it's better to control the RH in the same level in the future studies.

Relative humidity can influence the photochemical or aqueous-phase processing of SOA, it is stated that the aerosol liquid water content may show a linear increase as a function of RH (at RH > 60%) during the three seasons in Beijing, indicating the potential impacts of aqueous-phase processing at high RH levels (Xu et al., 2017). Therefore, RH 18-23% or RH 44-49% are both relatively low, and photochemical oxidation may still play the leading role in these two experiments. In the future, we really hope we could strictly control the temperature, RH, and other conditions to deeply investigate how the aerosol ages as a function of equivalent days of atmospheric oxidation, under certain gas-phase and heterogeneous oxidations.

According to the reviewers' comments, We have added the necessary description in the Method and Conclusion section as the following text shows:

"**Revised text in Method 2.3.2:** "The Go: PAM conditions for vehicle and cooking experiments could be seen in Tables 3 and Table 4, respectively. Their experiment conditions (such as residence time and RH ranges) were not completely the same because of the inherent difference and experimental design between two sources. Whereas,

some comparisons could be still analyzed in the similar OH exposure, and their RH conditions were both low where photochemical oxidations instead of aqueous-phase processing still dominated the chemical evolution process (Xu et al., 2017)."

"**Revised text in Conclusion:** "There are some uncertainties of our Go: PAM simulation. We focused more on the photochemical oxidation of SOA under low RH levels, but aqueous-phase processing at high RH levels may also have impacts to SOA production. In the future, it'll be better to strictly control the RH, high/low $NO_x$ or $SO_2$, additional inorganic seeds, and so forth, to deeply investigate how the aerosol ages as a function of equivalent days of atmospheric oxidation."

*References:*

*Xu, W., Han, T., Du, W., Wang, Q., Chen, C., Zhao, J., Zhang, Y., Li, J., Fu, P., Wang, Z., Worsnop, D. R., and Sun, Y.: Effects of Aqueous-Phase and Photochemical Processing on Secondary Organic Aerosol Formation and Evolution in Beijing, China, Environmental science & technology, 51, 762-770, 10.1021/acs.est.6b04498, 2017.*

*7.A lot of text in Sections 2.3.2 and S2 looks too similar. Some of it may be cut to reduce redundancy.*

Thanks for the reviewer's comment. Some of the contents in S2 have been cut to reduce redundancy.

*8.Line 98: a dash is needed between "Beijing" and "Chengde".*

Thanks for the correction. We have revised it to "Beijing-Chengde".

*9.Ling 196: "Hongkong" -> "Hong Kong".*

Thanks for the correction. We have revised it to "Hong Kong".

*References:*

*Liao, K., Chen, Q., Liu, Y., Li, Y. J., Lambe, A. T., Zhu, T., Huang, R.-J., Zheng, Y., Cheng, X., Miao, R., Huang, G., Khuzestani, R. B., and Jia, T.: Secondary Organic Aerosol Formation of Fleet Vehicle Emissions in China: Potential Seasonality of Spatial Distributions, Environ. Sci. Technol., 55, 7276–7286, 2021.*

*Pagonis, D., Krechmer, J. E., de Gouw, J., Jimenez, J. L., and Ziemann, P. J.: Effects of gas–wall partitioning in Teflon tubing and instrumentation on time-resolved measurements of gas-phase organic compounds, Atmos. Meas. Tech., 10, 4687–4696, 2017.*

---

## Author Response (AR3)

**Responses to Comments**

We appreciate the constructive comments from the reviewers on this manuscript. We have answered the reviewers' questions point to point in the following paragraphs (the texts italicized are the comments, the texts indented are the responses, and the texts in blue are revised parts in the new manuscript). In addition, all changes made are marked in the revised manuscript.

**I. Comments from Reviewer 2**

*1. In the methods and conclusions sections, please add a couple sentences referencing companion publications-in-preparation that discuss the speciated gas- and condensed-phase compounds and their individual SOA formation potentials.*

We appreciate the reviewer's comment. We have added necessary description referencing the companionate publications-in-preparation in the Conclusion section.

"**Revised text in Conclusion:** In another companionate publication-in-preparation (Song Kai, Guo Song*, *et al.,* Cooking emitted S/IVOCs are a large pool of SOA formation precursors, *In preparing*), gas and particle phase VOCs and S/IVOCs from four typical Chinese domestic cuisines are quantified. It is found that 26-78% of cooking SOA could be explained from the oxidation of VOCs and S/IVOCs. Moreover, oxygenated compounds were the most abundant in particle phase, including acids, furans, amides and esters. In contrast, significant differences were found in gas phase among four cuisines, for example, Kung Pao Chicken and shallow-frying Tofu showed larger proportion of aromatics."

Besides, the mass spectra that we obtained could be used in the ambient air, and we referred it briefly in the Conclusion section.

"**Revised text in Conclusion:** The contribution of vehicle SOA and cooking SOA for OA were estimated by ME-2 model in urban Beijing (Zhang Zirui, Hu Min*, et al. Secondary Organic Aerosol Formation from Urban Lifestyle Sources in Beijing. *In preparing*). It is found that cooking SOA (27-42% of OA) and vehicle SOA (58-73% of OA) presented different diurnal pattens, implying their different formation pathways. Similar features of urban lifestyle SOA were found between laboratory and field results."

**II. Comments from Reviewer 3**

*1. The authors have adequately addressed most of the reviewers' comments in this round of revision. However, I disagree with a point in their response to one of my comments (Response 3): High/Low NOx conditions should not be distinguished by VOC-to-NOx ratio, but by the fate of organic peroxy radicals (see e.g. Wennberg (2013)). Regardless of VOC concentrations, organic peroxy radicals react dominantly with NO as long as the latter's concentration is at ppb level. If the authors intend to show the variability of high/low NOx regimes in the ambient air in Beijing, it needs to be based on organic peroxy loss pathways.*

Thanks for the reviewer's constructive and professional comments. We agree with the reviewer. It's better to distinguish high/low $NO_x$ conditions by the fate of organic peroxy radical than VOCs-to-NOx ratio. Frankly, high-NOx pathways in organic peroxy radical chemistry may allow fragmentation of organics to occur at lower degree of oxidation than low-NOx pathways. However, the measurements of $RO_2$、VOCs or S/I VOCs components are usually uncomplete and full of uncertainties. Therefore, VOCs-to-NOx ratio were widely used to estimate SOA formation condition in laboratory simulation and field studies (Liu, et al., ACP, 2018). Even so, relatively simple laboratory results also have many uncertainties when it was applied in the ambient air, considering the extremely complicated ambient conditions.

In summary, the variability of high/low NOx regimes in the ambient air in Beijing were not described in the manuscript. This is actually one of uncertainties of our research. We greatly appreciate the reviewer's comments that have pointed out the uncertainty of our research. We have added the related description of many uncertainties in Conclusion method, and also the potential application of our laboratory results.

"**Revised text in Conclusion:** "In the future, it'll be better to strictly control the temperature, RH, **high/low $NO_x$** or $SO_2$, additional inorganic seeds, and so forth, to deeply investigate how the aerosol ages as a function of equivalent days of atmospheric oxidation.""

"**Revised text in Conclusion:** The contribution of vehicle SOA and cooking SOA for OA were estimated by ME-2 model in urban Beijing (Zhang Zirui, Hu Min*, et al. Secondary Organic Aerosol Formation from Urban Lifestyle Sources in Beijing. *In preparing*). It is found that cooking SOA (27-42% of OA) and vehicle SOA (58-73% of OA) presented different diurnal pattens, implying their different formation pathways. Similar features of urban lifestyle SOA were found between laboratory and field results."

References:

Liu, T., Wang, Z., Wang, X., and Chan, C. K.: Primary and secondary organic aerosol from heated cooking oil emissions, Atmospheric Chemistry and Physics, 18, 11363-11374, 10.5194/acp-18-11363-2018, 2018.

*2. Regarding Response 4, while it is fine to focus on the particle phase, the manuscript should not leave to readers an impression that cooking SOA is mainly from heterogeneous oxidation of POA unless the authors show the evidence.*

Thanks for the reviewer's helpful advice. We are sorry to cause this misunderstanding. The simulated SOA could be generated by the photochemical oxidation from gaseous precursors and the heterogeneous oxidation from POA. In this work, we couldn't clearly analyze the relative strength of gaseous photochemistry or that of the heterogeneous chemistry. We have added necessary expressions in the start of Discussion section and the uncertainty part of Conclusion section.

"**Revised text in Discussion 3.1:** The simulated SOA could be generated by the photochemical oxidation from gaseous precursors and the heterogeneous oxidation from POA."

"**Revised text in Conclusion:** Furthermore, the relative strength of the photochemical oxidation from gaseous precursors and the heterogeneous oxidation from POA were not deeply distinguished in this work."

*3. For Response 1, I think that Deming et al. (2020) is an appropriate reference for the discussion about stainless steel tubing loss/delay.*

Thanks for the careful review. We agree with the reviewer that it's important to evaluate the pipe loss in order to ensure the preciseness of experimental design. The reviewer has recommended an article for us (Deming et al., 2012). They expand recent results by comparing different types of Teflon and other polymer tubing, as well as glass, uncoated and coated stainless steel and aluminum, and other tubing materials by measuring the response to step increases and decreases in organic compound concentrations. It is found that conductive PFA tubing and silonite are the best choices for simultaneous gas and particle sampling. In our work, all stainless steel pipes are silanized, which fully meet the needs of particulate and gas sampling. In order to eliminate misunderstandings, we have added the word "silanized" before the stainless steel in the manuscript. In addition, we have added the evaluation of particle penetrating fraction according to the equivalent pipe length method in SI (Wiedensohler et al., 2012).

"**Revised text in Method section:** All the sampling tubes are made of silanized stainless steel which is appropriate for a simultaneous gas and particle sampling (Deming et al., 2019;Wiedensohler et al., 2012)**."**

"**Revised text in SI Section S1:** There were 5 meters long from the sampling port to the Go: PAM, the flow rate was 5.5 L/min (HR-Tof-AMS and other instruments jointly determined the flow rate), **and the penetrating fraction was more than 90% for those particles whose diameter was larger than 10 nm (equivalent pipe length method) (Wiedensohler et al., 2012).** There were 2 meters long from the Go: PAM to the HR-ToF-AMS, the flow rate was 1

L/min (HR-ToF-AMS and its drainage system determined the flow rate), **and the penetrating fraction was more than 88% for those particles whose diameter was larger than 10 nm (equivalent pipe length method) (Wiedensohler et al., 2012).**

References:

Wiedensohler, A., Birmili, W., Nowak, A., Sonntag, A., Weinhold, K., Merkel, M., Wehner, B., Tuch, T., Pfeifer, S., Fiebig, M., Fjäraa, A. M., Asmi, E., Sellegri, K., Depuy, R., Venzac, H., Villani, P., Laj, P., Aalto, P., Ogren, J. A., Swietlicki, E., Williams, P., Roldin, P., Quincey, P., Hüglin, C., Fierz-Schmidhauser, R., Gysel, M., Weingartner, E., Riccobono, F., Santos, S., Grüning, C., Faloon, K., Beddows, D., Harrison, R., Monahan, C., Jennings, S. G., O'Dowd, C. D., Marinoni, A., Horn, H. G., Keck, L., Jiang, J., Scheckman, J., McMurry, P. H., Deng, Z., Zhao, C. S., Moerman, M., Henzing, B., de Leeuw, G., Löschau, G., and Bastian, S.: Mobility particle size spectrometers: harmonization of technical standards and data structure to facilitate high quality long-term observations of atmospheric particle number size distributions, Atmospheric Measurement Techniques, 5, 657-685, 10.5194/amt-5-657-2012, 2012.

References given by Reviewer 3: Deming, B. L., Pagonis, D., Liu, X., Day, D. A., Talukdar, R., Krechmer, J. E., de Gouw, J. A., Jimenez, J. L., and Ziemann, P. J.: Measurements of delays of gas-phase compounds in a wide variety of tubing materials due to gas–wall interactions, Atmos. Meas. Tech., 12, 3453–3461, 2019.

Wennberg, P. O.: Let's abandon the "high NOx" and "low NOx" terminology, IGAC News, 50, 3–4, 2013.

---

## Author Response (AR4)

**Responses to Comments**

*The Editor's decision: Publish subject to technical corrections. "This looks good to go, although I'd recommend changing the word 'emission' to 'emissions' in the title."*

Thanks for the editor's patient and professional review for the last 7 months. We have revised "Cooking Emission" to "Cooking Emissions" in the title as well as in the main text of manuscript and supplement.